# Characterizing Emerging Canine H3 Influenza Viruses

**Luis Martinez-Sobrido**[1]*, **Pilar Blanco-Lobo**[1], **Laura Rodriguez**[1¤a], **Theresa Fitzgerald**[2], **Hanyuan Zhang**[3,4], **Phuong Nguyen**[2], **Christopher S. Anderson**[2], **Jeanne Holden-Wiltse**[5], **Sanjukta Bandyopadhyay**[5], **Aitor Nogales**[1], **Marta L. DeDiego**[2], **Brian R. Wasik**[6], **Benjamin L. Miller**[3,4], **Carole Henry**[7], **Patrick C. Wilson**[7], **Mark Y. Sangster**[2], **John J. Treanor**[2], **David J. Topham**[2], **Lauren Byrd-Leotis**[8,9], **David A. Steinhauer**[8], **Richard D. Cummings**[8,9], **Jasmina M. Luczo**[10], **Stephen M. Tompkins**[10], **Kaori Sakamoto**[11], **Cheryl A. Jones**[10], **John Steel**[8¤b], **Anice C. Lowen**[8], **Shamika Danzy**[8], **Hui Tao**[8], **Ashley L. Fink**[12], **Sabra L. Klein**[12], **Nicholas Wohlgemuth**[12], **Katherine J. Fenstermacher**[12], **Farah el Najjar**[12], **Andrew Pekosz**[12], **Lauren Sauer**[13], **Mitra K. Lewis**[13], **Kathryn Shaw-Saliba**[13], **Richard E. Rothman**[13], **Zhen-Ying Liu**[14], **Kuan-Fu Chen**[14], **Colin R. Parrish**[6], **Ian E. H. Voorhees**[6], **Yoshihiro Kawaoka**[15], **Gabriele Neumann**[15], **Shiho Chiba**[15], **Shufang Fan**[15], **Masato Hatta**[15], **Huihui Kong**[15], **Gongxun Zhong**[15], **Guojun Wang**[16,17], **Melissa B. Uccellini**[16,17], **Adolfo García-Sastre**[16,17,18,19], **Daniel R. Perez**[20], **Lucas M. Ferreri**[20], **Sander Herfst**[21], **Mathilde Richard**[21], **Ron Fouchier**[21], **David Burke**[22], **David Pattinson**[22], **Derek J. Smith**[22], **Victoria Meliopoulos**[23], **Pamela Freiden**[23], **Brandi Livingston**[23], **Bridgett Sharp**[23], **Sean Cherry**[23], **Juan Carlos Dib**[24], **Guohua Yang**[23], **Charles J. Russell**[23], **Subrata Barman**[23], **Richard J. Webby**[23], **Scott Krauss**[23], **Angela Danner**[23], **Karlie Woodard**[23], **Malik Peiris**[25], **R. A. P. M. Perera**[25], **M. C. W. Chan**[25], **Elena A. Govorkova**[23], **Bindumadhav M. Marathe**[23], **Philippe N. Q. Pascua**[23], **Gavin Smith**[26], **Yao-Tsun Li**[26], **Paul G. Thomas**[27], **Stacey Schultz-Cherry**[23]*

1 Department of Microbiology and Immunology, University of Rochester, Rochester, New York, United States of America, 2 David H. Smith Center for Vaccine Biology and Immunology, University of Rochester, Rochester, New York, United States of America, 3 Department of Dermatology, University of Rochester, Rochester, New York, United States of America, 4 Materials Science Program, University of Rochester, Rochester, New York, United States of America, 5 Department of Biostatistics and Computational Biology and Clinical and Translational Science Institute, University of Rochester, Rochester, New York, United States of America, 6 Baker Institute for Animal Health, Department of Microbiology and Immunology, College of Veterinary Medicine, Cornell University, Ithaca, New York, United States of America, 7 The Department of Medicine, Section of Rheumatology, The Knapp Center for Lupus and Immunology Research, The University of Chicago, Chicago, Illinois, United States of America, 8 Department of Microbiology and Immunology, Emory University School of Medicine, Atlanta, Georgia, United States of America, 9 Beth Israel Deaconess Medical Center, Department of Surgery and Harvard Medical School Center for Glycoscience, Harvard Medical School, Boston, Massachusetts, United States of America, 10 Center for Vaccines and Immunology, University of Georgia, Athens, Georgia, United States of America, 11 Department of Pathology, University of Georgia, Athens, Georgia, United States of America, 12 W. Harry Feinstone Department of Molecular Microbiology and Immunology, Johns Hopkins Bloomberg School of Public Health, Baltimore, Maryland, United States of America, 13 Department of Emergency Medicine, Johns Hopkins University School of Medicine, Baltimore, Maryland, United States of America, 14 Department of Emergency Medicine, Chang Gung Memorial Hospital, Taiwan, 15 Influenza Research Institute, Department of Pathobiological Sciences, School of Veterinary Medicine, University of Wisconsin-Madison. Madison, Wisconsin, United States of America, 16 Department of Microbiology, Icahn School of Medicine at Mount Sinai, New York, New York, United States of America, 17 Global Health and Emerging Pathogens Institute, Icahn School of Medicine at Mount Sinai, New York, New York, United States of America, 18 Department of Medicine, Division of Infectious Diseases, Icahn School of Medicine at Mount Sinai, New York, New York, United States of America, 19 The Tisch Cancer Institute, Icahn School of Medicine at Mount Sinai, New York, New York, United States of America, 20 Department of Population Health, University of Georgia, Athens, Georgia, United States of America, 21 Department of Viroscience, Erasmus MC, Rotterdam, The Netherlands, 22 Center for Pathogen Evolution, Department of Zoology, University of Cambridge, Cambridge, United Kingdom, 23 Department of Infectious Diseases, St. Jude Children's Research Hospital, Memphis,



**Data Availability Statement:** All relevant data are within the manuscript and its Supporting Information files.

**Funding:** This project was funded by the National Institute of Allergy and Infectious Diseases (NIAID),

National Institutes of Health (NIH), Department of Health and Human Services, Centers of Excellence for Influenza Research and Surveillance (CEIRS) contract Nos. HHSN272201400004C, HHSN272201400005C, HHSN272201400006C, HHSN272201400007C, HHSN272201400008C and NIH grant P41GM103694 to RDC. The funders had no role in study design, data collection and analysis, decision to publish, or preparation of the manuscript.

**Competing interests:** AG-S. is inventor of patents on influenza virus vaccines owned by the Icahn School for Medicine at Mount Sinai and licensed to Medimmune, BI Vetmedica, Vivaldi Biosciences, Zoetis and Avimex.

Tennessee, United States of America, **24** Tropical Health Foundation, Santa Marta, Magdalena, Colombia, **25** School of Public Health, Li Ka Shing Faculty of Medicine, The University of Hong Kong, Hong Kong Special Administrative Region, Republic of China, **26** Programme in Emerging Infectious Diseases, Duke-NUS Medical School, Singapore, **27** Department of Immunology, St. Jude Children's Research Hospital, Memphis, Tennessee, United States of America

¤a Current address: Agencia Española de Medicamentos y Productos Sanitarios, Madrid, Spain.
¤b Current address: Influenza Division, Centers for Disease Control and Prevention, Atlanta, Georgia, United States of America
* Luis_Martinez@URMC.Rochester.edu, l.martinez@txbiomed.org (LM); Stacey.Schultz-Cherry@STJUDE.ORG (SS)

## Abstract

The continual emergence of novel influenza A strains from non-human hosts requires constant vigilance and the need for ongoing research to identify strains that may pose a human public health risk. Since 1999, canine H3 influenza A viruses (CIVs) have caused many thousands or millions of respiratory infections in dogs in the United States. While no human infections with CIVs have been reported to date, these viruses could pose a zoonotic risk. In these studies, the National Institutes of Allergy and Infectious Diseases (NIAID) Centers of Excellence for Influenza Research and Surveillance (CEIRS) network collaboratively demonstrated that CIVs replicated in some primary human cells and transmitted effectively in mammalian models. While people born after 1970 had little or no pre-existing humoral immunity against CIVs, the viruses were sensitive to existing antivirals and we identified a panel of H3 cross-reactive human monoclonal antibodies (hmAbs) that could have prophylactic and/or therapeutic value. Our data predict these CIVs posed a low risk to humans. Importantly, we showed that the CEIRS network could work together to provide basic research information important for characterizing emerging influenza viruses, although there were valuable lessons learned.

### Author summary

The 2009 influenza pandemic was a stark reminder that ongoing vigilance is critical to protect the public from an influenza pandemic. The continual evolution of influenza viruses and emergence from animal reservoirs, leads to the need to quickly identify strains that pose a public health risk. In these studies, members of the National Institutes of Allergy and Infectious Diseases (NIAID) Centers for Excellence in Influenza Research and Surveillance (CEIRS) network worked together to demonstrate that the emerging canine H3 influenza viruses posed a low risk to public health and identified several therapeutic options in the event of an emergence. In addition to providing important new basic research, many lessons were learned that may be important in dealing with any emerging disease outbreak.

## Introduction

In the United States (US), the 2017–2018 influenza season was one of the most severe in recent memory resulting in a total of 171 laboratory-confirmed influenza-associated pediatric deaths

and over 30,000 confirmed hospitalizations [1]. Devastating influenza outbreaks are not unique to humans. In recent years, H3 subtype canine influenza viruses (CIVs) have resulted in many thousands or millions of respiratory infections in dogs in the United States.

The H3N2 CIV first appeared in the United States in Illinois in the spring of 2015 but was derived from a virus that had been circulating in China and South Korea since 2005 or 2006 [2–4]. Dogs infected with H3N2 CIV exhibited respiratory influenza-like symptoms, shed virus through nasal discharge, and the virus was readily transmitted between dogs via direct contact [5] causing several outbreaks throughout the United States [4,6]. This H3N2 CIV is avian in origin [2,4–9] but prior to 2015, a separate CIV strain derived from equine H3N8 influenza viruses, circulated in the United States [10].

The zoonotic potential of H3 CIV is unknown. Serological studies suggested that human H1N1 and H3N2 viruses could infect dogs, albeit inefficiently, and A/Hong Kong/68-like and influenza B viruses were isolated from dogs in Taiwan in 1971 [11,12]. Swine-origin H1N1 CIV have also been isolated in dogs in China [13]. Since dogs are susceptible to infection with at least three different influenza A subtypes (H3N8, H3N2 and H1N1) and can also be infected with diverse human influenza viruses, these animals have the potential to act as mixing vessels in which novel viruses are generated by reassortment. Genetic exchange through reassortment allows the formation of chimeric genotypes in which some gene segments are well adapted to human hosts. Compared to strains in which the full genome has evolved in a non-human host, such reassortant viruses may more easily overcome the species barrier to cause an outbreak in humans [14].

In the studies described here, members of the National Institutes of Allergy and Infectious Diseases (NIAID) Centers of Excellence for Influenza Research and Surveillance (CEIRS) network collaborated to provide important basic research to address the public health risk of emerging H3 CIVs. The network performed studies specifically addressing the criteria described in the public health algorithms developed by the Centers for Disease Control and Prevention (CDC) and the World Health Organization (WHO) to estimate the potential risk to human health and of pandemic emergence. Our studies demonstrated that the H3 CIVs bound preferentially to α2,3-linked sialic acids ("avian-like receptors") yet replicated in primary human nasal and bronchial epithelial cells. Mild-to-moderate disease was observed in infected mice depending on the mouse line. The viruses transmitted in ferrets and guinea pigs and were sensitive to matrix 2 (M2) and neuraminidase (NA) inhibitors. There were low levels of preexisting antibody (Ab) immunity in humans and vaccination with a recent human H3N2 strain did not lead to increased serum neutralizing antibody (NAb) titers to the H3N2 CIV. Finally, we identified nine anti-H3 human monoclonal antibodies (hmAbs), derived from influenza-vaccinated individuals, with strong neutralizing activity against both H3N2 and H3N8 CIVs. Together, our results suggest that H3 CIVs pose a low risk to humans, with younger people representing the highest population at risk. Importantly, we showed that the CEIRS network could work together to provide basic research information characterizing emerging influenza viruses and we identified areas where more efficient and rapid responses to future emerging influenza viruses can be developed. Lessons that can be applied to other human disease outbreaks.

## Results

### Risk assessment criteria

The goal of this exercise was to determine how effectively the CEIRS network could work together to provide key basic research on H3 CIVs addressing components of the 10 scientific criteria outlined in the CDC Influenza Risk Assessment Tool (IRAT, [15]) and/or WHO Tool

for Influenza Pandemic Risk Assessment (TIPRA, [16]). These criteria include specific properties of the virus (genomic variation, receptor binding, transmission in laboratory animals, and antiviral susceptibility or resistance), attributes of the population (pre-existing immunity, disease severity and pathogenesis, and antigenic relationship of the emerging virus to vaccines or vaccine candidates), as well as ecology and epidemiology (global distribution in animals, infection in animal species, and human infections (Table 1). This information can then be used to calculate risk using the IRAT/TIPRA algorithms.

## Viral strains

Phylogenetic analysis of canine H3 viral sequences obtained from the Influenza Research Database (fludb.org) was performed and several genetically related viral strains (Table 2) were chosen for comparison in this exercise. A/canine/Illinois/41915/2015 (CIV-41915) and the human seasonal A/Bethesda/55/2015 (Beth15) H3N2 viruses were selected because viral isolates were available within the network. A/canine/Indiana/1177-17-1/2017 (rCIV-1177) and A/canine/Illinois/11613/2015 (rCIV-11613) H3N2 viruses were chosen for full genome synthesis and recombinant viruses were generated using reverse genetics [17–19]. The ability to successfully synthesize an emerging virus is an important exercise as it is likely that in future outbreaks sequences will be obtained and shared before viral isolates are available [20]. Additional human H3N2 and a canine H3N8 virus (Table 1) were used in the human immunity studies to

**Table 1. Summary of risk assessment scientific criteria.**

| *Risk Elements for Likelihood Risk Score* | *Representative Criteria (examples)* |
|---|---|
| Receptor binding properties | • α2,3-linked sialic acids<br>• α2,3 and α2,6-linked sialic acids<br>• α2,6-linked sialic acids |
| Transmission in animal models | • No direct or respiratory droplet<br>• Consistent direct contact<br>• Consistent respiratory droplet |
| Genomic characteristics | • No molecular signatures of human infection and disease.<br>• Molecular signatures<br>• Contains genes derived from a virus circulating in mammals |
| Infection in animals | • Wild species<br>• Poultry or mammals<br>• Endemicity<br>• Numbers of impacted animals |
| Geographic distribution in animals | • Local and contained<br>• Regional within well-defined geographic boundaries.<br>• Widespread no boundaries |
| *Risk Elements for Impact Risk Score* | |
| Disease severity (human) | • Uncomplicated<br>• Uncomplicated but some severe illness (underlying conditions)<br>• Severe disease |
| Population immunity | • Cross-reactive antibodies > 30% of population except children < 17<br>• Cross-reactive antibodies > 30% of people >50<br>• Minimal (<10%) cross-reactive antibodies |
| Susceptibility to antiviral treatment | • Resistant to adamantanes<br>• Above + resistant to 1 NA inhibitor<br>• Above + resistant to > 1 NA inhibitor |
| Genomic characteristics | As above |
| Receptor binding properties | As above |
| Human infection | • Single case<br>• Isolated or sporadic cases<br>• Frequent separate clusters |

**Table 2. Viruses used in the study and their abbreviations.**

| Subtype | Virus Abbreviation | Virus Name |
|---|---|---|
| H3N2 | CIV-41915[a] | A/canine/Illinois/41915/2015 |
| H3N2 | rCIV-1177[b] | A/canine/Indiana/1177-17-1/2017 |
| H3N2 | rCIV-11613[b] | A/canine/Illinois/11613/2015 |
| H3N2 | rCIV-11613-mCherry[b] | A/canine/Illinois/11613/2015 mCherry |
| H3N2 | Beth15[a] | A/Bethesda/55/2015 |
| H3N2 | HK99[a] | A/Hong Kong/1174/1999 |
| H3N2 | WY03[a] | A/Wyoming/3/2003 |
| H3N2 | rWY03[b] | A/Wyoming/3/2003 |
| H3N2 | rWY03-mCherry[b] | A/Wyoming/3/2003 mCherry |
| H3N2 | WI05[a] | A/Wisconsin/67/2005 |
| H3N2 | HK14[a] | A/Hong Kong/4801/2014 |
| H3N8 | CIV-23[a] | A/canine/NY/dog23/2009 |
| H3N8 | rCIV-23[b] | A/canine/NY/dog23/2009 |
| H3N8 | rCIV-23-mCherry[b] | A/canine/NY/dog23/2009 mCherry |

[a]Natural isolate
[b]recombinant virus

more broadly assess responses to H3 viruses. Viral stocks and synthetic viruses were distributed to the network participants within weeks of the project initiation. Viral passage history is included in the Methods section.

**Replication in primary human respiratory epithelial cells and explants.** To monitor replication potential in humans, primary human nasal and bronchial epithelial cells (hNECs and hBECs, respectively) were cultured at an air-liquid interface [21,22] and then inoculated with Beth15, CIV-41915 and rCIV-1177 H3N2 viruses at a multiplicity of infection (MOI) of 0.1 (Fig 1). Viral replication was assessed at 32˚C and 37˚C to determine whether the physiological range of temperatures of the respiratory tract influenced virus replication. Beth15 and CIV-41915 viruses had similar kinetics and reached similar peak infectious virus titers of around $10^6$ tissue culture infectious dose 50 ($TCID_{50}$)/ml in hNECs at 32˚C. In contrast, rCIV-1177 virus had slower kinetics and lower peak infectious virus titers of $10^5$ $TCID_{50}$/ml (Fig 1A). At 37˚C, the Beth15 and rCIV-1177 viruses replicated with similar kinetics and peak titers, but CIV-41915 virus had higher peak infectious virus titers (Fig 1A). In hBECs, at 32˚C the CIVs replicated to significantly lower peak titers compared to Beth15 virus (Fig 1B) while there were no differences in replication of the three viruses in hBEC cultures at 37˚C (Fig 1B). In contrast, only the human Beth15 and A/Hong Kong/1174/99 (HK99 H3N2) viruses productively replicated in *ex vivo* human bronchus explant cultures, with the CIVs showing no replication (Fig 1C). These results indicate that the CIVs replicate efficiently in primary human respiratory cells but not bronchus explant cultures.

## Receptor binding and HA stability

Both receptor binding specificity and HA stability at acidic pH are important criteria for assessing emergence risk, as those have been observed to be key determinants in other examples of successful adaptation and transmission when crossing species barriers [23]. The amino acids around the receptor binding site of the H3 CIVs suggest preferential binding to α2,3-linked sialic acid (SA) receptors like other Eurasian lineage avian H3 viruses [3]. Indeed, glycan binding analysis confirmed that the binding profile of CIV-41915 differed qualitatively

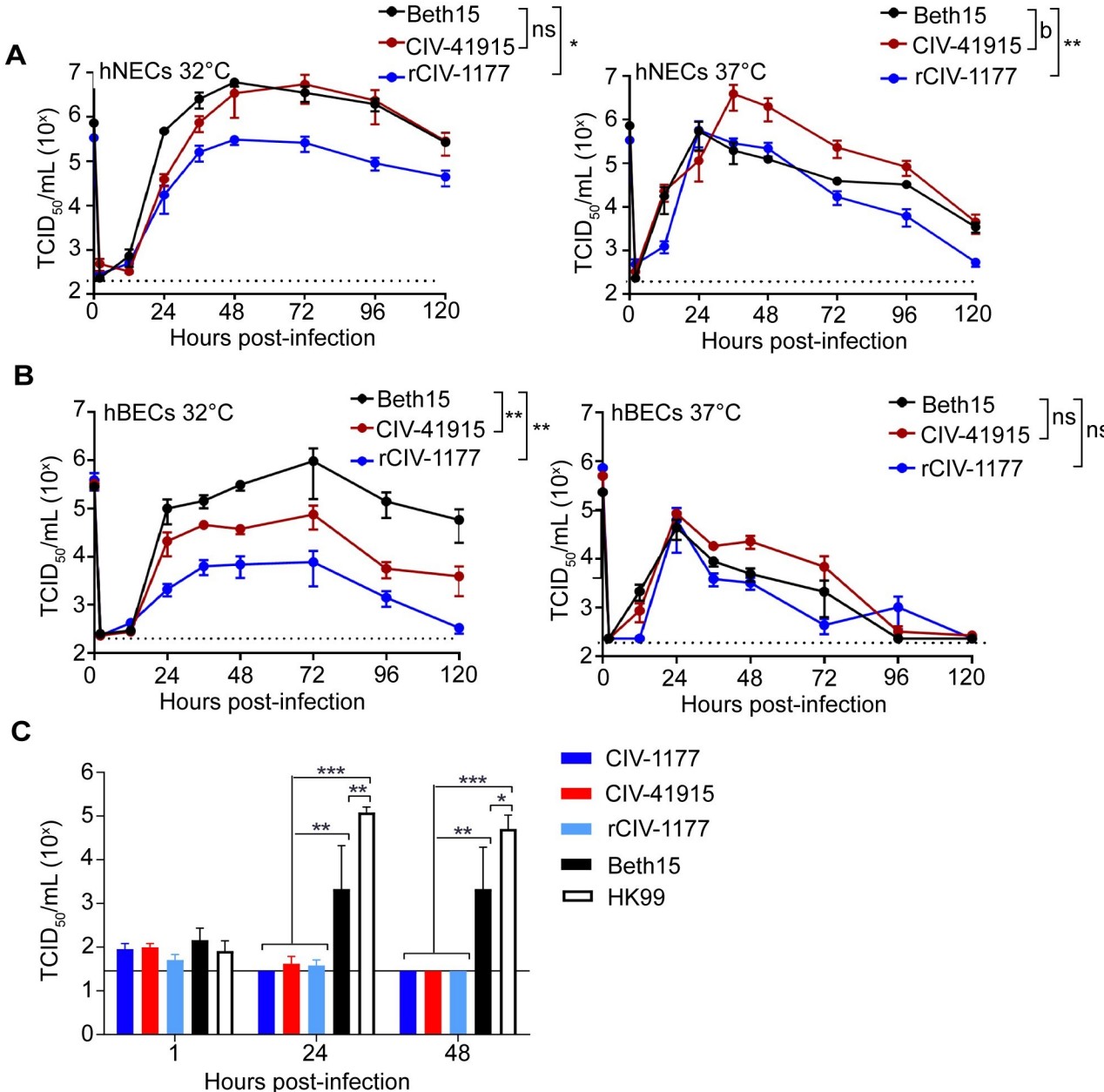

**Fig 1. Replication in primary cells and explants.** (A) hNEC or (B) hBEC cultures were inoculated with Beth15, CIV-41915 or rCIV-1177 viruses at a MOI of 0.1 or MOI of 1 and incubated at 32°C or 37°C. At the indicated time, apical media was collected, and virus titers determined. Data are pooled from 2 independent experiments with n = 3 wells per virus for each experiment (n = 6 total). Two-way ANOVA was used for statistical analysis (a = p<0.05, b = p<0.001 compared to Beth15 virus). Dotted line indicates limit of detection. (C) Human bronchus explant culture was submerged in $10^6$ TCID$_{50}$/ml virus for 1 hour at 37°C, washed and placed onto a surgical sponge in a 24-well tissue culture plate filled with 1 ml/well of culture medium to create an ALI. Supernatant was collected at 1, 24, and 48 hpi and virus titer determined. Experiments were performed with tissues from 3 donors (n = 3). Two-way ANOVA was used for statistical analysis (* = p<0.03, ** = p<0.0005, *** = p<0.0001 compared to mock).

from that of the human Beth15 H3N2 virus, as the latter virus preferentially bound α2,6-linked SA receptors, while CIV-41915 preferentially bound avian α2,3-linked SA receptor (Fig 2).

HA acid stability is the pH at which HA is triggered to undergo conformational changes needed to trigger fusion of the viral envelope with the endosomal membrane, or in the absence of a target membrane the pH at which virion infectivity is irreversibly inactivated. HA stability

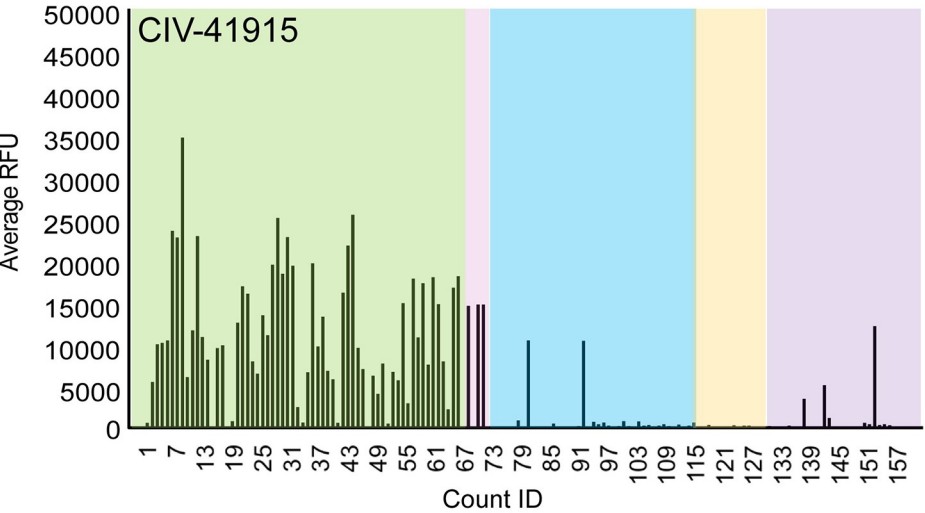

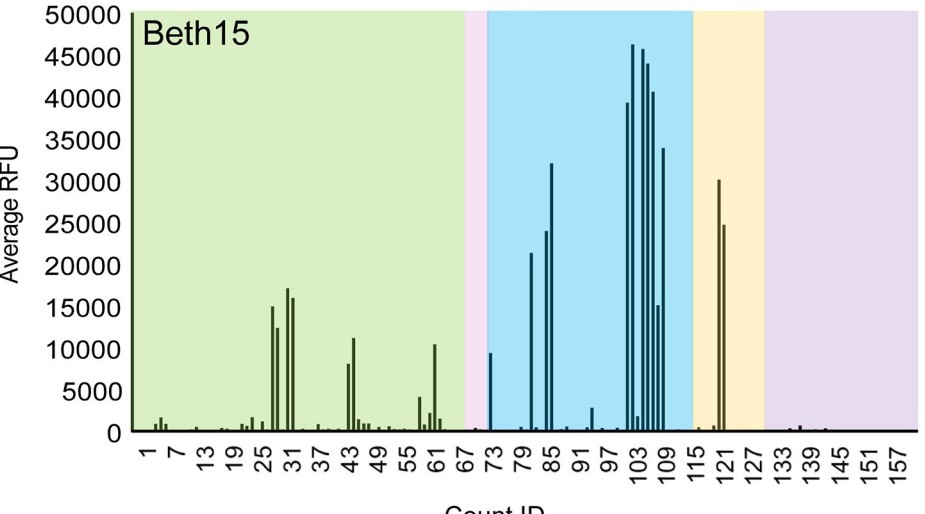

**Fig 2. Glycan array binding.** Fluorescently labeled CIV-41915 and Beth15 viruses were incubated on the glycan microarray for 1 hour at 4°C, to inhibit viral neuraminidase activity, then the slide was washed to remove unbound virus and scanned using a ProScanArray microarray scanner for Alexa Fluor 488 fluorescence and results shown as RFU. Each bar represents a single glycan. Green box = α2,3; pink box = α2,3 + α2,6; blue box = α2,6; orange box = α2,8; and purple box = miscellaneous + NeuGc glycans.

has been linked to pandemic potential and the ability to cross the species barrier, suggesting that it is an important viral characteristic to measure when assessing risk [24]. The H3 CIVs and the human H3N2 viruses had similar pH of fusion values as measured by syncytia formation (5.45–5.50, Table 3). For human H3N2, the pH values of HA-mediated fusion and

**Table 3. HA acid stability of H3N2 human and CIVs.**

| Virus | pH of syncytia | pH of inactivation |
|---|---|---|
| Beth15 | 5.45 | 5.55 |
| CIV-41915 | 5.50 | 5.09 |
| rCIV-1177 | 5.45 | <5.2 |

inactivation were within 0.1 units. However, for the H3N2 CIVs the inactivation pH values were approximately 0.3–0.4 units lower than their activation pH values, showing these viruses had increased resistance to acid inactivation. Despite the divergence of HA activation and inactivation pH values of the H3N2 canine viruses, the values remained within the range of those reported for human-adapted influenza viruses. Overall, these studies suggest that while the H3N2 CIV maintains avian receptor binding specificity, HA stability of CIVs resemble that of mammalian viruses.

## Pathogenicity in mice

Viruses that bind to α2,3-linked SA receptors typically do not require adaptation to cause disease in mice [25]. Indeed, intranasal inoculation of DBA/2J mice (n = 5 per group) with $10^6$ $TCID_{50}$ CIV-41915 or rCIV-1177 led to significant weight loss and 60% (CIV-41915) to 100% (rCIV-1177) mortality. Euthanasia upon reaching humane end points accounted for 60% of DBA/2J mortality following infection with rCIV-1177 (Fig 3A and 3B). In contrast, BALB/c (Fig 3A and 3B) and C57Bl/6 mice exhibited no significant morbidity (weight loss) during infection despite having lung viral titers. However, DBA/2J mice had ~1-log higher titers compared to BALB/c at days 3 and 5 post-infection with the CIV-41915 virus and at 5 dpi with the rCIV-1177 virus (Fig 3C). Detection of increased viral loads, and increased susceptibility of DBA/2J mice to influenza virus infection is consistent with previous findings [26,27]. Beth15 virus caused no morbidity and lacked detectable viral titers in the lungs. These data highlight that unlike the human H3N2 virus, the H3N2 CIVs replicate in mice but morbidity was only observed with specific mouse lines.

## Transmission in ferrets and guinea pigs

A key property of emerging influenza strains is their ability to infect and transmit in mammals. To assess transmission of CIV-41915 virus, donor ferrets were intranasally inoculated with $10^6$ $TCID_{50}$/ml, while guinea pigs were inoculated with between 10 and 1,000 plaque forming units (pfu). Twenty-four-hour post-infection (hpi), naïve animals were placed in direct (both hosts) or airborne contact (ferrets only) with donor (inoculated) animals. Animals were monitored for clinical signs and viral titers in nasal washes measured through 7–8 dpi (Fig 4). None of the infected animals exhibited clinical signs of infection including weight loss or fever, but the virus did transmit to 100% of the direct contact ferrets (Fig 4A). It also transmitted to 73 to 83% of the contact guinea pigs regardless of the viral dose (Fig 4B and Table 4). While there was no airborne transmission with rCIV-1177 virus, one CIV-41915 respiratory contact ferret had nasal titers at day 7 post-infection and seroconverted with an hemagglutination inhibition (HAI) titer of 320. These studies highlight that the CIVs can transmit by direct contact, but that respiratory droplet transmission is inefficient [3,9]. It is important to note that the titer used to infect ferrets, while the accepted dose, may not be representative of the infectious dose in nature.

## Antiviral susceptibility

The antiviral susceptibility of a new virus is a major determinant when assessing the impact an emerging strain might have on the human population. Thus, the mean inhibitory concentration 50 ($IC_{50}$) values of neuraminidase (NA) inhibitors (Oseltamivir, Zanamir and Peramivir) and the M2 inhibitor Amantadine were determined against CIV-41915, rCIV-1177 and Beth15 viruses by NA enzyme inhibition and phenotypic assays respectively [28]. The H3 CIVs were sensitive to all Food and Drug Administration (FDA)-approved antiviral drugs in contrast to the Beth15 virus, which was resistant to amantadine as predicted by the presence of

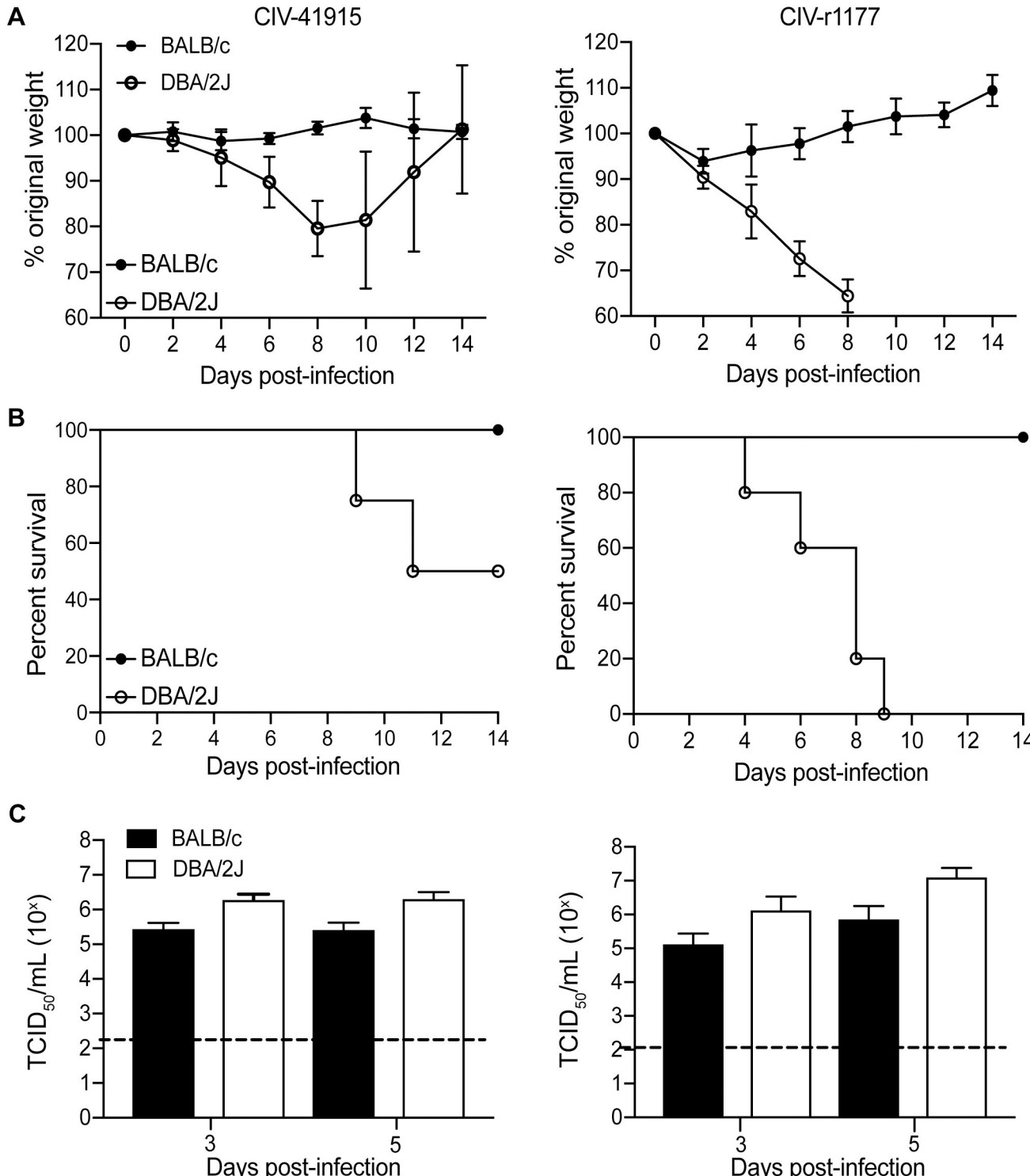

**Fig 3. Pathogenicity in mice.** (A-B) Body weight and survival: Six-week-old BALB/c and DBA/2J mice (n = 5 per group) were lightly anesthetized with isoflurane and inoculated intranasally with $10^6$ $TCID_{50}$ CIV-41915 (left panel) or rCIV-1177 (right panel) in 50 μl DMEM. Mice were monitored daily for clinical disease and weighed every other day. (C) Virus loads present in whole lungs were determined 3 and 5 dpi (n = 5 mice per group). Virus titers are expressed as mean±95% CI and * = p <0.05 as compared to BALB/C.

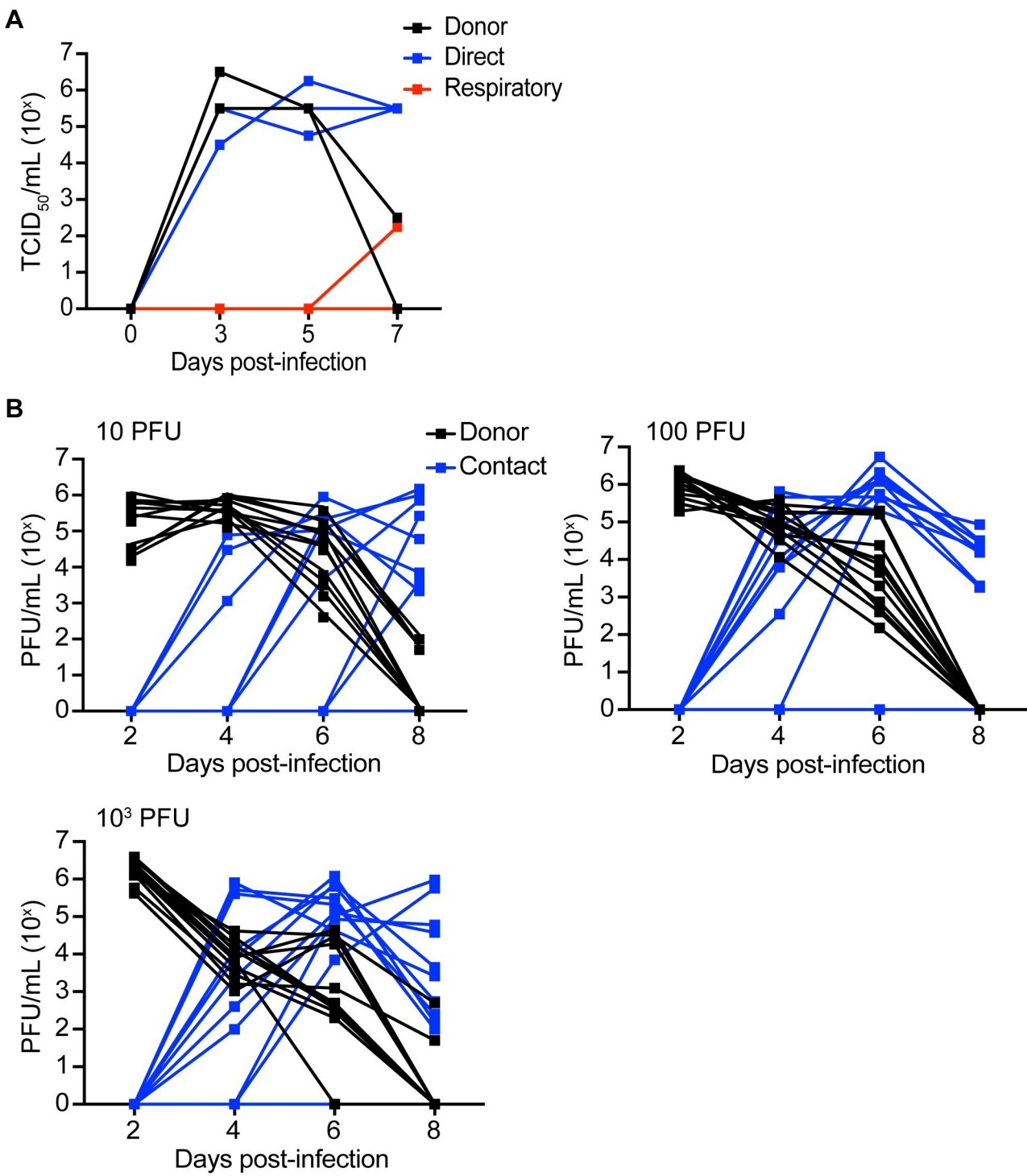

**Fig 4. Transmission in ferrets and guinea pigs.** (A) 4 to 6-month-old seronegative ferrets were lightly anesthetized and intranasally inoculated with $10^6$ $TCID_{50}$ of CIV-41915 in 1.0 ml sterile PBS. At 1 dpi, naïve contacts were placed in the same cage with directly inoculated ferrets while respiratory contacts were placed in adjacent cages separated by a wire grill. Nasal washes were collected every other day for 7 dpi and viral titers determined in MDCK cells. Lines represent individual animals. (B) Guinea pigs were intranasally inoculated 10 to $10^3$ PFU. At 1 dpi, naïve contacts were placed in the same cage with directly inoculated animals. Nasal washes were collected from anesthetized animals on days 2, 4, 6 and 8 post-inoculation and viral titers determined in MDCK cells. Lines represent individual animals.

**Table 4. Infection and transmission of CIV-41915 in guinea pigs.**

| Inoculum (pfu) | Infected (%) | Transmissions (%) |
|---|---|---|
| 1 | 2/12 (17%) | 1/2 (50%) |
| 10 | 11/12 (92%) | 8/11 (73%) |
| 100 | 12/12 (100%) | 9/12 (75%) |
| 1000 | 12/12 (100%) | 10/12 (83%) |

the S31N substitution in M2 [29] (Table 5). These studies confirm that the H3 CIVs are sensitive to both FDA-approved antiviral drugs.

## Determining human immunity against CIVs

**Sequence-based antigenic distance calculations.** The level of population immunity is a major determinant when assessing the impact an emerging strain would have on the human population. This can be determined genetically by using the protein sequences of the HA and NA to predict antigenic relationships between virus and vaccine strains by sequence-based antigenic distance calculations [30–32] or using viral-specific antisera to perform antigenic cartography. Given the lack of ferret antisera to CIVs, we performed sequence-based antigenic distance calculations by analyzing 19,070 H3 sequences from 1963–2017 belonging to 10 animal species and 9 influenza subtypes, 21,792 N2 sequences from 1957–2017 and 2,290 N8 sequences from 1963–2017 belonging to 14/16 subtypes and 12/9 animal species respectively. Sequences were aligned and the number of amino acid changes between strains calculated (Hamming distance). H3 had an aligned length of 579 amino acids with a mean distance of 43.5 and a maximum Hamming distance of 357 giving maximum percent difference of 62%. N2 had an aligned length of 490 amino acids with a mean distance of 48, a maximum distance of 154, and a maximum percent difference of 31%. N8 had an aligned length of 470 amino acids with a mean distance of 40.3, a maximum distance of 113, and a maximum percent difference of 24%. Overall, the three proteins had similar mean Hamming distances, despite differences in aligned amino acid length; with HA having the highest variability between strains, while N2 and N8 had similar numbers of amino acid changes.

To facilitate visualization of the data sets, we employed classic multidimensional scaling (principal component analysis) to reduce the dimensionality of the Hamming distance matrix and allow "mapping" of the strains in 2 dimensions. H3 (Fig 5A) and N8 (Fig 5C) had similar conservation of distance, with N2 (Fig 5B) slightly reduced (GOF: 0.69, 0.69, 0.58, respectively). All proteins showed clear clustering with N8 forming 3 distinct clusters (Fig 5C). The H3 HA of both H3N2 CIV-11613 (designated IL15 in Fig 5A and 5B) and H3N8 CIV A/

**Table 5. Susceptibility of H3N2 CIVs to antiviral drugs in phenotypic assays.**

| Influenza virus | NA enzyme inhibition assay (Mean IC$_{50}$ ± SD, nM)[a] | | | Phenotypic assay (Mean IC$_{50}$ ± SD, µM)[b] |
|---|---|---|---|---|
| | Oseltamivir carboxylate | Zanamivir | Peramivir | Amantadine |
| CIV-41915 | 0.13 ± 0.01 | 0.85 ± 0.01 | 0.15 ± 0.02 | 0.6 ± 0.1 |
| rCIV-1177 | 0.20 ± 0.01 | 1.14 ± 0.03 | 0.19 ± 0.02 | 0.3 ± 0.1 |
| Beth15 | 0.16 ± 0.01 | 0.33 ± 0.02 | 0.12 ± 0.02 | > 100 |

[a]Concentration of NA inhibitor that reduced NA activity by 50% relative to a reaction mixture containing virus but no inhibitor. Values are the mean ± SD from three independent experiments.

[b]Performed in accordance with the CDC protocol. Values are the mean ±SD from three independent experiments.

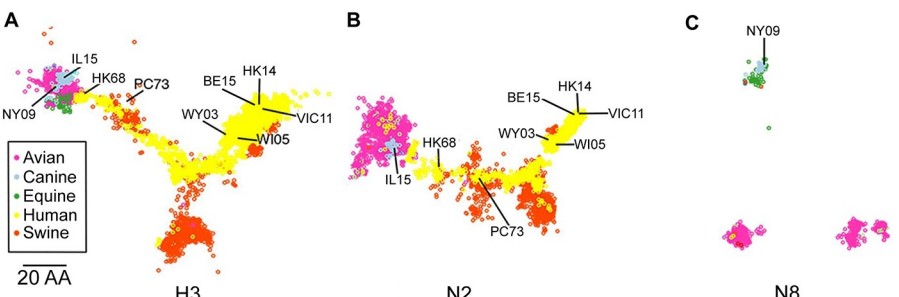

**Fig 5. Cartography mapping of H3, N2 and N8 influenza proteins.** Sequence-based maps of H3 (A), N2 (B) and N8 (C) viral proteins. Each point represents a single virus strain. Points are colored based on the species the virus was isolated from (A-C). Position of CIV-11613 (IL15), CIV-23 (NY09), HK68, PC73, Wy03, WI05, Vic11 and HK14 are indicated. Scale bar, 20 amino acids distance.

canine/NY/dog23/2009 (rCIV-23 but designated as NY09 in Fig 5A and 5C) clustered closely with avian and equine HA sequences (Fig 5A), likely reflecting their species of origin before moving into dogs. Likewise, clustering for N2 (Fig 5B) was close to primarily avian and to some human N2 strains. Overall, host species was the best predictor of clustering.

The relative location in sequence space can be used to predict antigenic relatedness to vaccine viral strains [32]. Thus, we determined the proximity of the viruses used in the H3N2 human vaccines to CIVs. We found that H3 CIV were closest to early H3N2 vaccine strains A/Hong Kong/1/1968 H3N2 (HK68) and A/Port Chalmers/1/1973 H3N2 (PC73) (Fig 5A). Similarly, the N2 of CIV-41915 was closely related to HK68 and PC73 vaccine strains (Fig 5B). In contrast, recent H3N2 vaccine strains such as A/Victoria/361/2011 (VI11) or A/Hong Kong/4801/2014 (HK14) resided furthest from H3N2 CIV strains (Fig 5A and 5B) suggesting that H3N2 CIVs are antigenically more like early human H3N2 viruses than contemporary strains.

**Pre-existing immunity in human sera.** Based on the sequence relationships observed, we hypothesized that sera from individuals exposed to early H3N2 influenza strains, either by infection or vaccination, would cross-react to H3 CIVs, but that little cross-reactivity would be found in sera from individuals exposed to more recent human H3N2 strains. We therefore examined the viral-specific IgG titer by enzyme-linked immunosorbent assay (ELISA), HAI, and NA-specific enzyme-linked lectin assay (ELLA) in 153 sera from healthy subjects born in the US between 1934 to 2012, and HAI titers in 225 sera from Guangzhou, China and 53 sera from Colombia, South America. Greater than 97% of the samples from the US cohort had antibodies against the human and canine H3 viruses by ELISA (IgG titer > 20) regardless of age (Fig 6A). In contrast, only 19.6% of the subjects had a detectable HAI titer ($\geq$10) against rCIV-11613 and 9.1% against the rCIV-23 H3N8 with only 7.8% and 0.65% of these people having seroprotective titers ($\geq$40 HAI [33]) respectively. This is compared to 95.4% of the people having HAI titer ($\geq$10) against human A/Wyoming/3/2003 (rWY03, Fig 6B) with 90.1% having seroprotective HAI titers $\geq$40 (Fig 6B). Interestingly, 93.3% of people with H3N2 CIV HAI titers were born before 1970 while 57.1% with HAI titer against H3N8 CIV were born before 1970. A similar trend was observed with the Guangzhou samples but not the Colombian samples where no people were HAI positive for H3 CIV (S1 Table).

Since NA antibodies can also provide protection [34,35], we tested for the presence of CIV N2 and N8 antibodies by ELLA (Fig 6C). 83% of the subjects had a NA inhibition titer (NAI titer $\geq$16) against rCIV-11613 N2, while only 49% of the subjects were positive against rCIV-23 N8. Most (98.7%) of the human sera contained functional anti-NA antibodies against human WY03 H3N2. Interestingly, the median NAI titer against rCIV-11613 N2 in subjects born before 1970 was significantly higher than those in younger subjects (median NAI titer

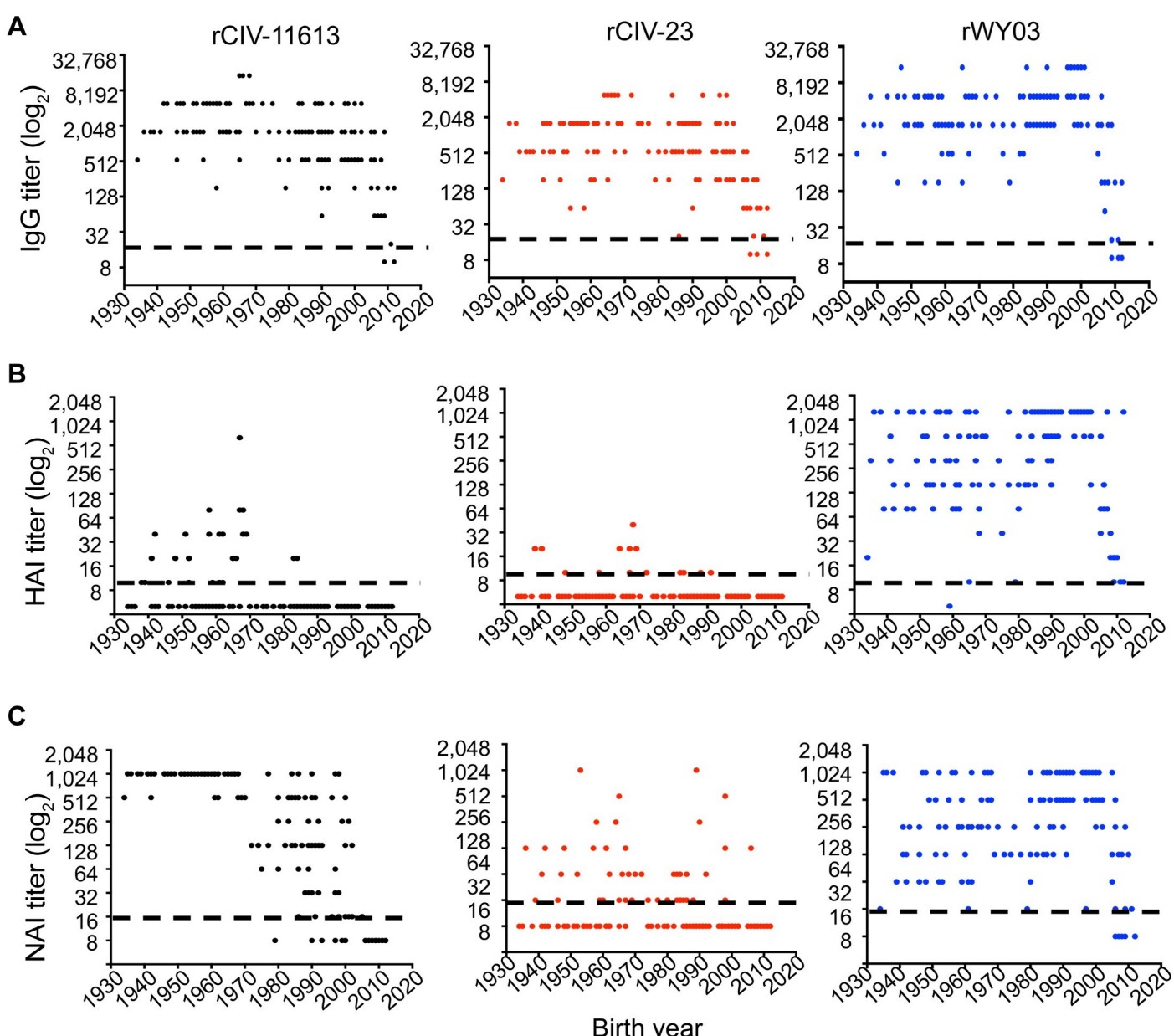

**Fig 6. Pre-existing population immunity against H3N2 and H3N8 CIV in human sera.** The presence of antibodies against rCIV-11613 H3N2 (black, left), rCIV-23 H3N8 (red, middle) and WY03 H3N2 (blue, right) was examined in triplicate by (A) ELISA, (B) HAI and (C) ELLA assays using 153 human sera samples collected from healthy subjects born between 1934 and 2012 and grouped in 10 years intervals. Dotted black lines indicate the limit of detection of each of the assays. Undetectable titers were assigned a value of 10, 5 and 8 for ELISA, HAI and NAI respectively. Each dot represents the mean titer (represented as $\log_2$) of a specific human subject.

1024 vs. 64; p<0.001). No association was found between the age and NAI titer against CIV-23 N8 (p = 0.07) or human rWY03 H3N2 virus (p = 0.08). Overall, antibodies recognizing canine N8 and N2 CIV proteins were found more frequently in people born before 1970. Younger people may therefore be considered at higher risk for H3 CIV infections while older people might have been exposed to influenza strains antigenically like currently circulating H3 CIVs, providing them with some residual cross-reactive and protective immunity.

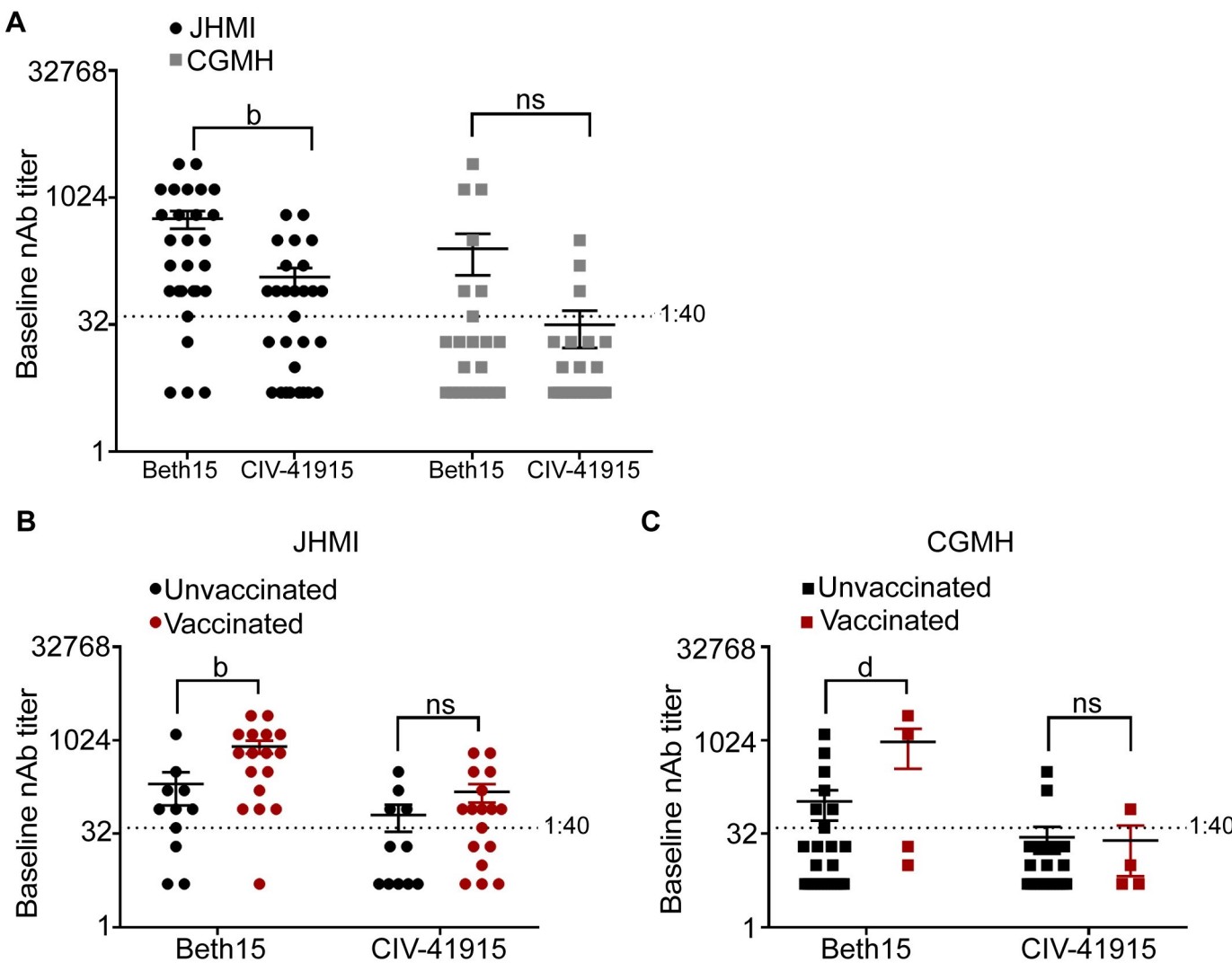

**Fig 7. Neutralizing Ab titers to human Beth15 and CIV-41915 H3N2 viruses in patients with confirmed H3N2 infections during the 2016–17 season.** (A) Human sera collected from Johns Hopkins Health System (JHMI) and Chang Gung Memorial Hospital (CGMH) in Taipei, Taiwan were subject to two-fold serial dilutions prior to incubation with 100 TCID$_{50}$ of either Beth 15 or CIV-41915. MDCK cells were then infected with the virus-sera mixture and following 24-hour incubation, cells were fixed and stained with Napthol blue black. The baseline neutralizing antibody titer was determined as the highest serum dilution that eliminated cytopathic effects in 50% of the wells. (B and C) Sera was collected from JHMI and CGMH at the time of hospital confirmation of H3N2 infection, with documentation about whether patients had or had not been vaccinated during the 2016–17 season and neutralizing antibody titers were determined as in (A). (b = p<0.001, d = p<0.0001 compared to Beth15).

Finally, we investigated the impact of confirmed human H3 infection with or without documented receipt of the seasonal influenza vaccine on CIV neutralizing Ab (NAb) titers. Serum samples were obtained from patients enrolled at either the Johns Hopkins Health System (JHMI) or Chang Gung Memorial Hospital (CGMH) in Taipei, Taiwan with confirmed H3N2 infection and documentation of receipt of the seasonal influenza vaccine during the 2016–17 season (Fig 7) [36,37]. At baseline (i.e., at the time of enrollment), seroprotective NAbs (>1:40) against CIV-41915 were detected in 54% in contrast to 81% against human Beth15 H3N2 virus (Fig 7A). At CGMH, the overall NAb titers were lower for both human Beth15 CIV-41915 H3N2 viruses when compared to the titers at JHMI but there was no statistically significant difference between Beth15 and CIV-41915 NAb titers (Fig 7A). As predicted by

antigenic cartography, vaccination with a contemporary HK14 H3N2 vaccine had no impact on NAb titers against CIV-41915 at either site but did increase titers against the human Beth15 H3N2 virus (Fig 7B and 7C). These results demonstrate that vaccination with human seasonal H3N2 vaccines does not change the Ab response to the H3 CIV in patients infected with H3N2 viruses.

## Protecting naïve populations

**Identifying human derived hmAbs cross-reactive to H3 CIVs.**   Our studies suggest that older people might have been exposed to influenza strains with some antigenic similarity to currently circulating H3 CIVs affording them with residual cross-reactive and protective immunity. Arrayed Imaging Reflectometry (AIR) has been used to provide information about anti-HA antibody titers in sera against different HA subtypes [38]. More recently, it has been used to demonstrate the multiplex capability of antibody arrays for live influenza virus characterization [39]. Here, we use AIR as a high-throughput screening approach to evaluate the cross-reactivity of a panel of recombinant hmAbs cloned from plasmablasts of volunteers vaccinated with different H3N2 strains (A/Perth/16/2009, A/Wisconsin/67/2005, A/Texas/50/2012, and A/Uruguay/716/2007) [39] against H3N2 and H3N8 CIVs and antigenically similar (HK68) or distinct (A/Wisconsin/67/2005, WI05) human H3N2 viruses. The array included a panel of seventeen H1-reactive, 65 H3-reactive and 6 H1/H3 cross-reactive hmAbs (Fig 8A). H7-reactive hmAbs were also included as controls for the human viruses (Fig 8B). Fig 9A–9D shows the data from representative arrays. We used a hierarchical clustering heatmap to reflect the quantitative response of each hmAb in the array to each of the indicated viruses (Fig 9E).

We identified 21 hmAbs reacting to both H3 CIVs, whereas 8 hmAbs reacted only to H3N2 rCIV-11613 (042-10065-2B03, 037-10036-5A01, C1-2A02, 229-1C01, 229-2E06, 008-10053-5E04, 009-10061-2E02, and 019-10117-3F02) and 2 hmAbs (C1-B01 and 228-3C06) only reacted to H3N8 rCIV-23. Of the 31 hmAbs that reacted to either H3 CIVs, 27 were also reactive to HK68 with only 15 reactive to WI05 H3N2 human viruses (Fig 9A–9D). Visualizing the AIR data as a cluster heatmap to examine the relationships between H1, H3, and H1/H3-reactive hmAbs demonstrated that H3 CIVs clustered with HK68 while WI05 clustered independently (Fig 9E), correlating with the sequence-based antigenic distance calculations map (Fig 5).

The cross-reactivity of 30 of the identified hmAbs (228-3C06 was no longer available) against H3 CIVs was further confirmed by immunofluorescent microscopy using mCherry-labeled viruses (S1 Fig). rWY03 and HK68 H3N2 human viruses served as internal controls. The fluorescence signal was quantified and relative binding of each hmAb was calculated considering the signal of the hmAb 017–10116 5B03 as the maximum 100% signal (Fig 10). The percentage of reactivity was used to categorize the hmAbs in 6 groups (Figs 9 and 10): Group 1: ≥30% reactivity against H3 CIVs and WI03 H3N2; Group 2: ≥30% reactivity against H3 CIVs, but <30% for r/WY03 H3N2; Group 3: ≥30% reactivity against H3N2 CIV-41915 but <30% for H3N8 rCIV-23 and WI03 H3N2; Group 4: ≥30% reactivity against H3N2 CIV-41915 and WI03 H3N2 but <30% for H3N8 rCIV-23; Group 5: ≥30% reactivity against WI03 H3N2 but <30% for rCIV-23 H3N8 and H3N2 CIV-41915; and, Group 6: non-reactivity or <30% reactivity against rCIV-23 H3N8 and/or H3N2 CIV-41915, or WI03 H3N2. Unfortunately, the epitopes of these hmAbs are unknown.

Overall, there was good correlation between the AIR and the immunofluorescence results for CIVs (summarized in Table 6). However, discordant results were found in Groups 3, 4 and 5, where some clones detected the CIVs by AIR but not by immunofluorescence, suggesting that the sensitivity of AIR might be greater than immunofluorescence. This analysis shows that

**A**

| Human-IgG | FITC-400 | FITC-200 | FITC-100 | Bovine-IgG |
|---|---|---|---|---|
| AO042610-4A01 | SA121509-1A06 | SA121509-1E04 | SA121509-2G03 | SA121509-3B03 |
| PW051310-4C06 | SA10066-1B03 | SA10066-1F03 | 018-10120-4D03 | 018-10120-4D06 |
| SFV009-2A06 | 042-10065-2B03 | 030-121509-3B01 | 037-10036-5A01 | 062860p153-E05 |
| BSA | 030-121509-2B03 | 047-050310-1G05 | SFV005-2G02 | 042-100809-2F04 |
| S6-B01 | 045-051310-2B06 | 045-051310-2C01 | 045-051310-1G05 | C1-B01 |
| C1-C02 | C1-D05 | C1-2A02 | C1-2.2F04 | C3-2.3C01 |
| C3-2.3F02 | C6-2.2E05 | C2-2.2G05 | C2-3.G05 | C1-3.7C02 |
| 034-100809-3E01 | 041-102909-3G01 | 229-1C01 | 229-1G01 | 229-2F03 |
| 229-2E06 | 229-2G06 | 228-3C06 | 235-1B06 | 235-1G01 |
| FITC-100 | FITC-200 | Bovine-IgG | FITC-400 | 042-10065-2B03 |
| 008-10053-1G05 | 008-10053-5E04 | 008-10053-5G01 | 008-10053-6C05 | 009-10061-2E02 |
| 009-10061-1D04 | 009-10061-2A05 | 009-10061-2C01 | 009-10061-2C06 | 009-10061-3B06 |
| 009-10061-2F03 | 011-10069-2C01 | 011-10069-3E06 | 011-10069-3G01 | 011-10069-5C01 |
| 011-10069-5D01 | 011-10069-5G01 | 011-10069-5G04 | 011-10069-2E06 | 011-10069-3D03 |
| 013-10078-3G01 | 014-10076-1E03 | 017-10116-5B03 | 017-10116-5D02 | 017-10116-5G02 |
| 019-10117-1B02 | 019-10117-3A06 | 019-10117-3C06 | 019-10117-3F01 | 019-10117-3F02 |
| 024-10128-3C04 | 024-10128-3F04 | 028-10134-4F03 | 030-121509-3B01 | 034-10040-1F01 |
| 034-10040-4C01 | 034-10040-4E01 | 034-10040-4F02 | 037-10036-5A01 | 041-10047-1C04 |
| SL10068-5F06 | SL10068-5G02 | SL10068-5G05 | TS050310-1F05 | TS050310-4E01 |
| Bovine-IgG | FITC-100 | FITC-200 | FITC-400 | Human-IgG |

Controls — H3-reactive mAbs — H1-reactive mAbs — H1/H3-reactive mAbs

**B**

| 13001-41-d105-1C03 | 13001-41-d105-2B03 | 13001-41-d105-2D01 | 13001-41-d105-5D06 | 13001-41-d105-5E04 |
|---|---|---|---|---|
| 3DR-7F02 | 042-10065-2B03 | 030-121509-3B01 | 037-10036-5A01 | 062860p153-E05 |
| SFV019-4E03 | 030-121509-2B03 | 047-050310-1G05 | SFV005-2G02 | 042-100809-2F04 |
| S6-B01 | 045-051310-2B06 | 045-051310-2C01 | 045-051310-1G05 | C1-B01 |
| C1-C02 | C1-D05 | C1-2A02 | C1-2.2F04 | C3-2.3C01 |
| C3-2.3F02 | C6-2.2E05 | C2-2.2G05 | C2-3.G05 | C1-3.7C02 |
| 034-100809-3E01 | 041-102909-3G01 | 229-1C01 | 229-1G01 | 229-2F03 |
| 229-2E06 | 229-2G06 | 228-3C06 | 235-1B06 | 235-1G01 |
| 042-10065-2B03 | 100-FITC | 200-FITC | 400-FITC | Human-IgG |
| 008-10053-1G05 | 008-10053-5E04 | 008-10053-5G01 | 008-10053-6C05 | 009-10061-2E02 |
| 009-10061-1D04 | 009-10061-2A05 | 009-10061-2C01 | 009-10061-2C06 | 009-10061-3B06 |
| 009-10061-2F03 | 011-10069-2C01 | 011-10069-3E06 | 011-10069-3G01 | 011-10069-5C01 |
| 011-10069-5D01 | 011-10069-5G01 | 011-10069-5G04 | 011-10069-2E06 | 011-10069-3D03 |
| 013-10078-3G01 | 014-10076-1E03 | 017-10116-5B03 | 017-10116-5D02 | 017-10116-5G02 |
| 019-10117-1B02 | 019-10117-3A06 | 019-10117-3C06 | 019-10117-3F01 | 019-10117-3F02 |
| 024-10128-3C04 | 024-10128-3F04 | 028-10134-4F03 | 030-121509-3B01 | 034-10040-1F01 |
| 034-10040-4C01 | 034-10040-4E01 | 034-10040-4F02 | 037-10036-5A01 | 041-10047-1C04 |

Controls — H1-reactive mAbs — H1/H3-reactive mAbs — H7-reactive mAbs — H3-reactive mAbs

**Fig 8. Layout information of the array.** (A) Array for rCIV-11613, rCIV-23, and human HK68 or (B) Wis05 H3N2 viruses. Human IgG was used as control for nonspecific IgG binding. Three anti-fluorescein (anti-FITC) solutions were uses as negative control at a concentration of 100 μg/ml (FITC-100), 200μg/ml (FITC-200) and 400μg/ml (FITC-400). Bovine IgG secondary antibodies reactive to the bovine sera used for blocking were used as positive control.

AIR could be used as a fast, reliable and sensitive high-throughput screening approach to identify hmAbs with cross-reactivity against different viruses, including H3N8 and H3N2 CIVs. Our data also indicate that previous vaccination or exposure to closely related human H3N2

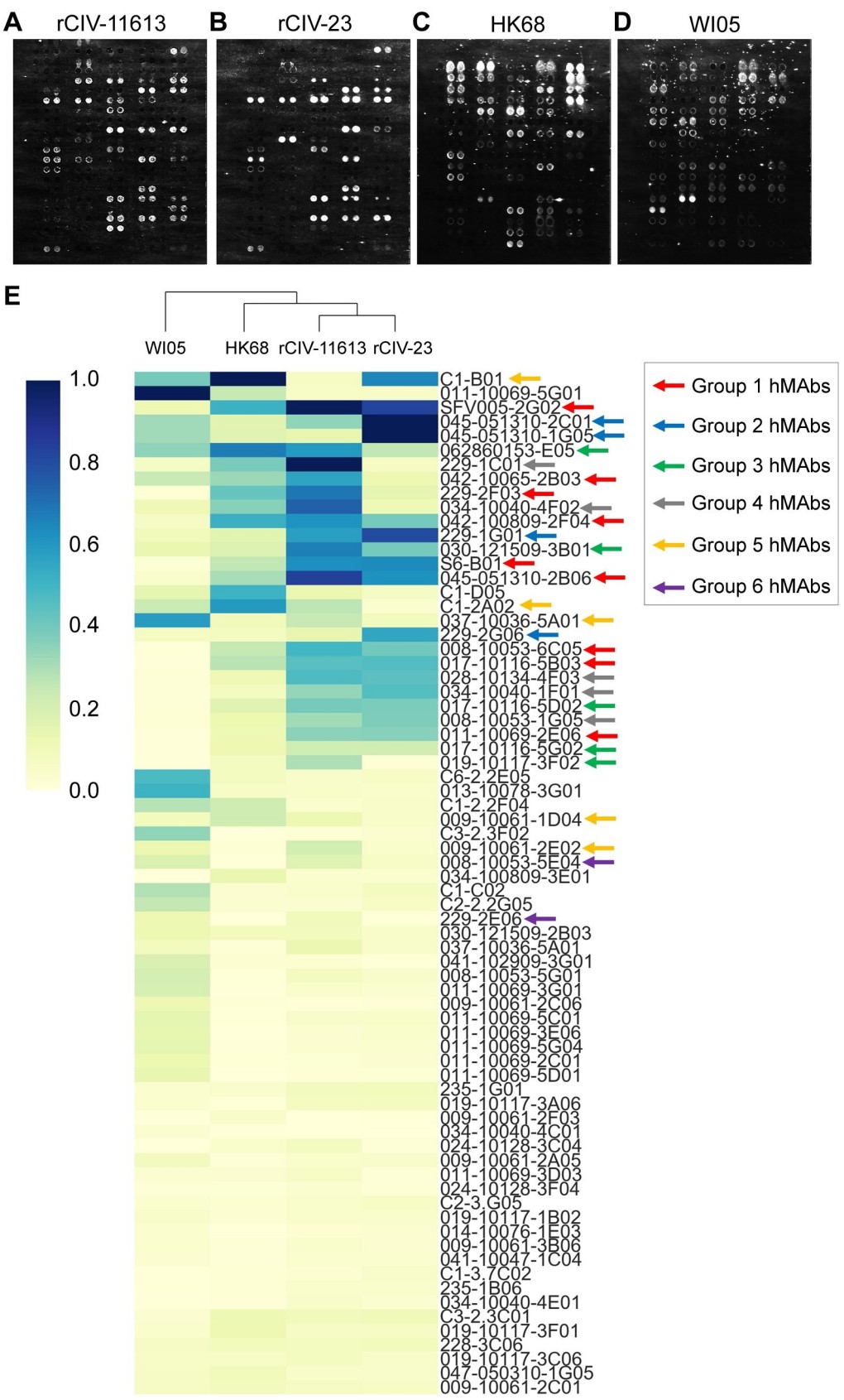

**Fig 9. Cross-reactivity of hmAbs by AIR.** AIR images of arrays exposed to (A) rCIV-11613 H3N2, (B) rCIV-23 H3N8, (C) HK68 and (D) WI05 viruses. (E) Hierarchical cluster map of the hmAb responses. Values in this map are scaled relative to the antibody producing the strongest response for each virus. Arrows show the antibody clustering groups.

viruses may elicit Ab responses that cross-react efficiently to the HA protein of H3N2 CIV and, to a lesser extent, H3N8 CIV.

## H3 cross-reactive hmAbs neutralize H3 CIVs in vitro

Passive immunotherapy could be a plausible strategy to control CIV infection [40,41]. We therefore used recombinant mCherry-expressing human (rWY03-mCherry) and canine

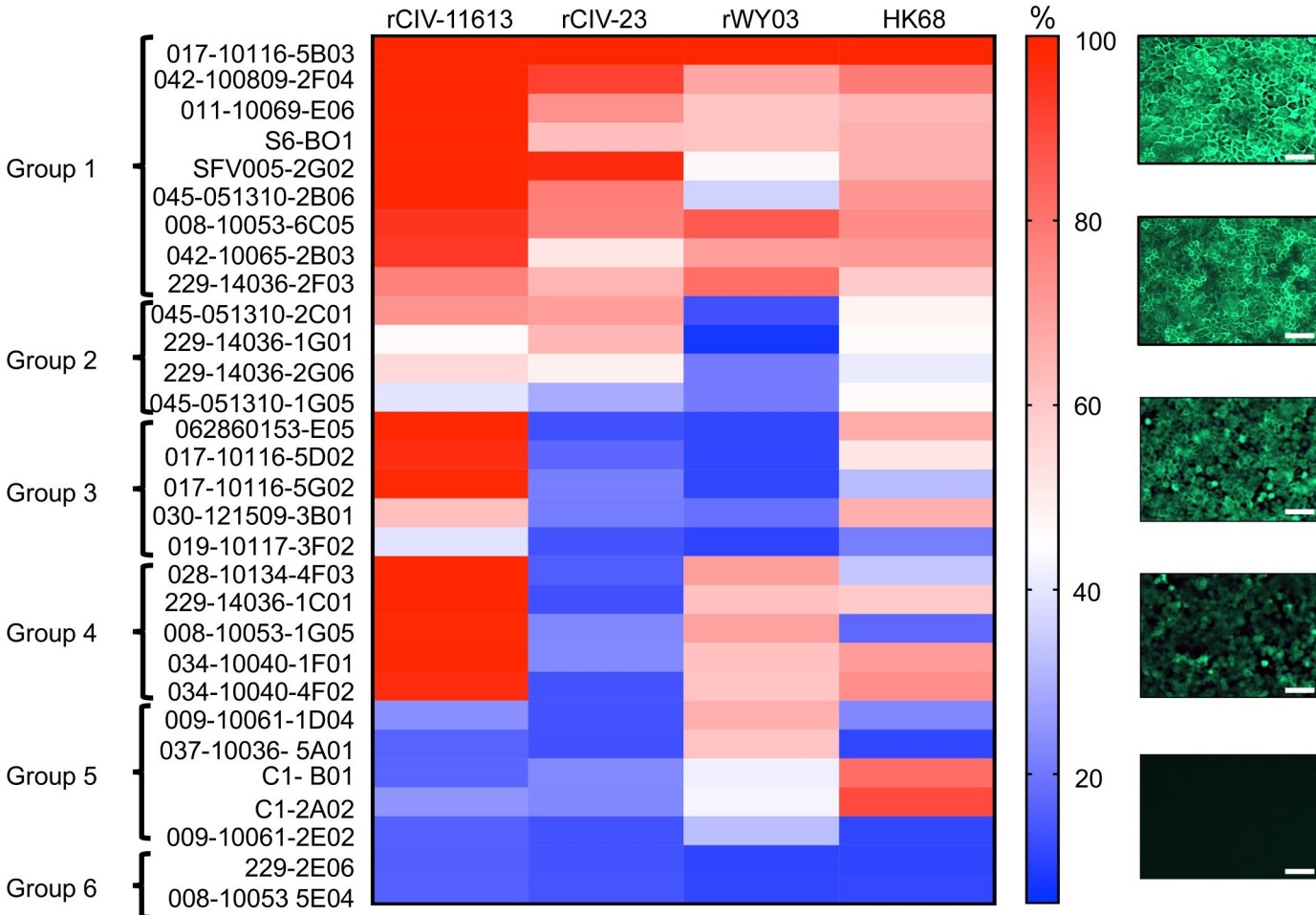

**Fig 10. Cross-reactivity of hmAbs by immunofluorescence.** MDCK cells were infected with rCIV-11613, rCIV-23, rWY03 or HK68 viruses at a MOI of 3. At 18 hpi, cells were fixed, permeabilized and incubated with 1 μg/ml of the indicated hmAbs. After incubation with a secondary anti-human FITC-conjugated antibody, fluorescence was imaged under a fluorescent microscope. Fluorescence intensity was measured using the ImageJ 1.51s and data was displayed using a Heatmap visualization method. For each virus, the hmAb providing the highest intensity (017–10116 5B03) was considered as 100% and was used to normalize the percentage of reactivity of the rest of the hmAbs. The percentage of reactivity was used to categorized the hmAbs in 6 groups: Group 1: ≥30% reactivity against H3 CIVs and rWY03 H3N2; Group 2: ≥30% reactivity against H3 CIVs, but <30% for rWY03 H3N2; Group 3: ≥30% reactivity against H3N2 rCIV-11613 CIV but <30% for H3N8 rCIV-23 and rWY03 H3N2; Group 4: ≥30% reactivity against H3N2 rCIV-11613 and rWY03 H3N2 but <30% for H3N8 rCIV-23 CIV; Group 5: ≥30% reactivity against rWY03 H3N2 but <30% for rCIV-23 H3N8 and H3N2 rCIV-11613; and, Group 6: non-reactivity or <30% reactivity against rCIV-23 H3N8 and/or H3N2 rCIV-11613, or rWY03. Representative images of the reactivity of the hmAbs against infected cells are shown in the right. Scale represent % of recognition. Scale bars, 200 μm.

**Table 6. Summary of the characteristics of the hmAbs.**

| | MAb | AIR | | IFA | | | $NT_{50}$ | | | HAI | | |
|---|---|---|---|---|---|---|---|---|---|---|---|---|
| | | rCIV H3N2 | rCIV H3N8 | rCIV H3N2 | rCIV H3N8 | r/Wy H3N2 | rCIV H3N2 | rCIV H3N8 | r/Wy H3N2 | rCIV H3N2 | rCIV H3N8 | r/Wy H3N2 |
| Group 1 | 017-10116-5B03 | + | + | + | + | + | 0.011 | 0.021 | 0.017 | >1 | >1 | 0.062 |
| | 042-100809-2F04 | + | + | + | + | + | 0.017 | 0.005 | 0.046 | >1 | >1 | >1 |
| | 011-10069-3E06 | + | + | + | + | + | 0.019 | 0.02 | 0.002 | >1 | >1 | >1 |
| | S6-B01 | + | + | + | + | + | 0.014 | 0.007 | 0.005 | >1 | >1 | >1 |
| | SFV005-2G02 | + | + | + | + | + | 0.090 | 0.017 | 1.56 | >1 | >1 | >1 |
| | 045-051310-2B06 | + | + | + | + | + | 0.058 | 0.015 | >2 | >1 | >1 | >1 |
| | 008-10053-6C05 | + | + | + | + | + | 0.044 | 0.033 | 0.004 | >1 | >1 | 0.25 |
| | 042-10065-2B03 | + | + | + | + | + | 0.076 | 0.49 | 0.014 | >1 | >1 | >1 |
| | 229–14036-2F03 | + | + | + | + | + | 0.043 | >2 | 0.012 | >1 | >1 | >1 |
| Group 2 | 045-051310-2C01 | + | + | + | + | - | >2 | 0.039 | >2 | >1 | >1 | >1 |
| | 229–14036-1G01 | + | + | + | + | - | 0.520 | >2 | >2 | >1 | >1 | >1 |
| | 229–14036 2G06 | + | + | + | + | - | 0.5 | 0.5 | >2 | >1 | >1 | >1 |
| Group 3 | 062860153-E05 | + | - | + | - | - | 0.063 | >2 | >2 | 0.015 | >1 | >1 |
| | 017-10116-5D02 | + | + | + | - | - | 0.004 | >2 | >2 | 0.125 | >1 | >1 |
| | 017-10116-5G02 | + | - | + | - | - | 0.041 | >2 | >2 | 0.5 | >1 | >1 |
| | 030-121509-3B01 | + | - | + | - | - | 0.6 | >2 | >2 | >1 | >1 | >1 |
| Group 4 | 028-10134-4F03 | + | + | + | - | + | 0.003 | >2 | 0.003 | >1 | >1 | 0.015 |
| | 229–14036-1C01 | + | - | + | - | + | 0.005 | >2 | 0.018 | >1 | >1 | 0.062 |
| | 008-10053-1G05 | + | + | + | - | + | 0.004 | 0.015 | 0.001 | >1 | >1 | 0.007 |
| | 034-10040-1F01 | + | + | + | - | + | >2 | >2 | 0.014 | >1 | >1 | 0.031 |
| | 034-10040-4F02 | + | - | + | - | + | 0.009 | >2 | 0.028 | >1 | >1 | 0.015 |
| Group 6 | 229-2E06 | - | - | - | - | - | >2 | >2 | >2 | >1 | >1 | >1 |
| | 008-10053-5E04 | - | - | - | - | - | >2 | >2 | >2 | >1 | >1 | >1 |

(rCIV-11613-mCherry and rCIV-21-mCherry) H3N2 viruses to examine the ability of the identified cross-reactive hmAbs to neutralize H3 CIV infections *in vitro* using a fluorescence-based microneutralization assay [42,43]. Eight of 9 hmAbs from Group 1 (Fig 11A) and 1 of 5 from Group 4 neutralized both H3N2 and H3N8 CIVs (Fig 11D). In contrast, 1 of 9 from Group 1 (Fig 11A), 3 of 4 from Group 3 (Fig 11C) and 3 from Group 4 (Fig 11D) specifically neutralized only the H3N2 rCIV-11613-mCherry while only 1 of 3 from Group 2 specifically neutralized H3N8 rCIV-23-mCherry (Fig 11B). Only hmAbs from Groups 1 and 4 displayed neutralization activity against human H3N2 rWY03-mCherry virus (Fig 11A and 11D), and none of the hmAbs from Group 6 displayed neutralization activity (Fig 11E). HAI assay, which generally only detects antibodies that bind the head domain of HA [44], identified only 3 hmAbs from Group 3 with activity against H3N2 rCIV-11613 (Table 6). No HAI titer was detected for the other 18 hmAbs at 0.01 mg/ml, the limit of detection in our assay. Overall, these results show that plasmablast cells from influenza vaccinated or infected humans could be used as sources for hmAbs that recognize and functionally neutralize H3 CIVs.

## Discussion

Identification of emerging influenza strains and assessing their pandemic potential is a crucial component of surveillance (https://www.hhs.gov/about/agencies/oga/global-health-security/pandemic-influenza/index.html; [45,46]). Reassortment and zoonotic transmission are the most likely threats, and the causative viruses must be quickly identified and characterized so

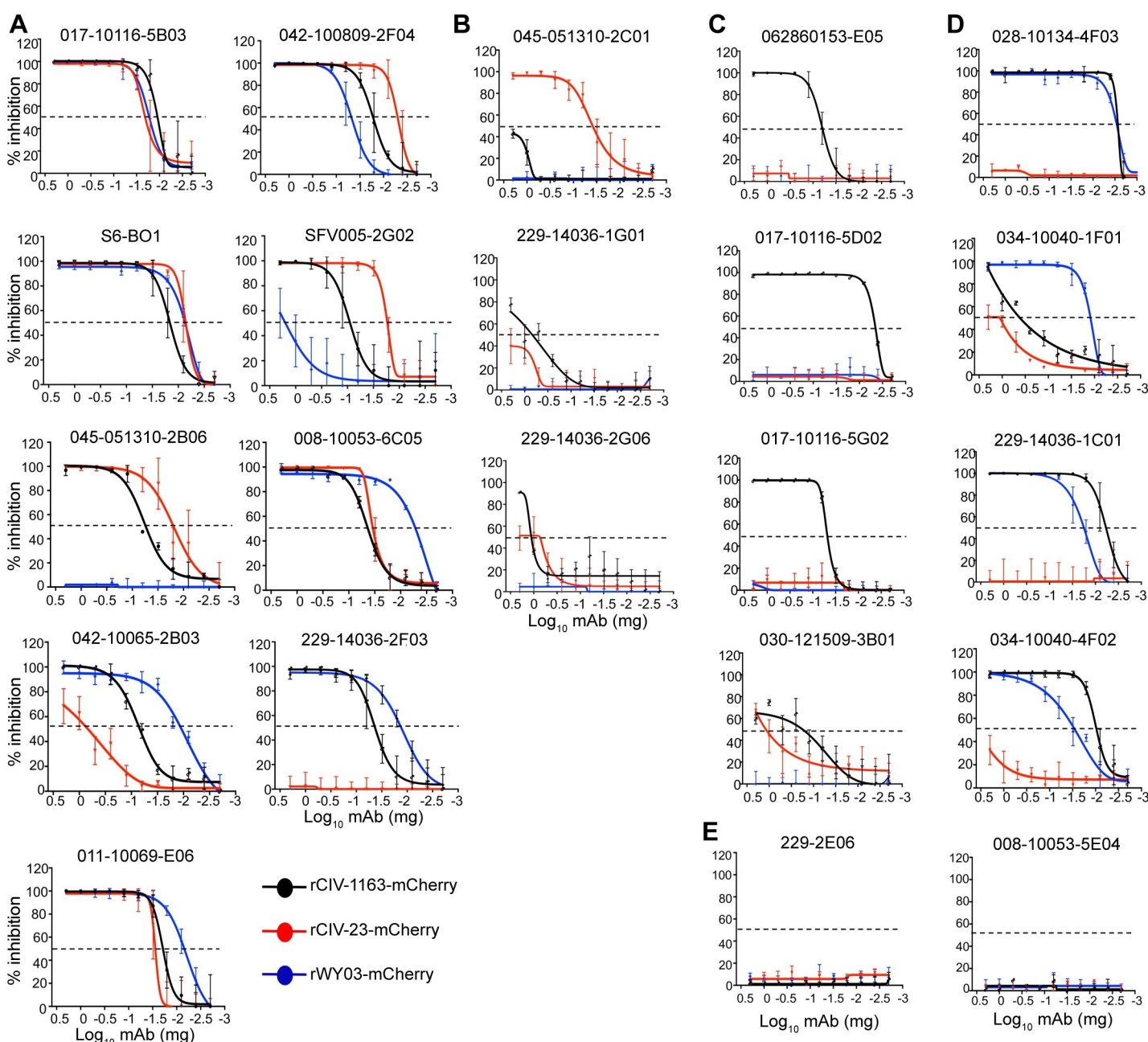

**Fig 11. Neutralizing activity of hmAbs.** MDCK cells were infected with the mCherry-expressing rCIV-23 H3N2 (black line), rCIV-23 H3N8 (red line) or rWY03 H3N2 (blue line) recombinant viruses at 0.005 MOI. After 1-hour adsorption at room temperature, cells were incubated with 2-fold serial dilutions of the indicated hmAbs (starting concentration of 0.2 mg/ml) belonging to the Groups 1 (A), 2 (B), 3 (C), 4 (D) or 6 (E). Group 5 hmAbs ($\geq$30% reactivity against rWY03 H3N2 but <30% for rCIV-23 H3N8 and rCIV-23 H3N2) were not included in the assay. At 60 hpi, fluorescence expression was quantified using a fluorescence plate reader and sigmoidal dose response curves were used to calculate the hmAbs concentration that reduced the virus infectivity by 50% (Neutralization titer 50, $NT_{50}$). Mock-infected and infected cells in the absence of hmAbs were used as controls to normalize the percentage of inhibition. Data show the mean and +/- SDs of the results determined for triplicate. An inhibition of 50% is indicated as a dotted black line.

that control and preventative measures can be taken. For example in 1997, the human H5N1 isolate was collected from the trachea of a three-year-old child and rapidly distributed to WHO Collaborating Centers for analysis [47–49]. Studies concluded that while that H5N1 isolate was not easily transmissible, it should be closely monitored since reassortment could

potentially result in a devastating pandemic [49]. Similarly, a zoonotic outbreak of H7N9 virus in 2013 precipitated a rapid assessment of viral pathogenicity and transmissibility in model animals [50].

The 2017–2018 influenza season brought problems with responses to H3N2 subtype seasonal influenza strains to the forefront of public health. Decreased vaccine efficacy and increased complications leading to hospitalizations [51,52] highlight the need to have carefully planned and preemptive response to both existing seasonal and new influenza pandemics. The continual emergence of novel influenza strains from non-human hosts, as well as new antigenic and virulence variants from humans, puts immense pressure on public health agencies to determine which of the many viruses in circulation may pose a risk to public health. Making well informed decisions about which viruses to focus on is also a key issue, given the cost of producing and stockpiling a candidate vaccine virus or developing other countermeasures. To aid in the decision process, WHO and CDC developed related algorithms that can be populated by specific information including (but not limited to) specific features of the virus, disease and spread and population immunity (Table 1) to predict risk [15,16]. A summary of influenza A viruses assessed using the CDC IRAT can be found at https://www.cdc.gov/flu/pandemic-resources/monitoring/irat-virus-summaries.htm. The goal of this exercise was to determine how quickly the CEIRS network could provide critical data on a new virus.

The global CEIRS network is unique in its ability to do the broad basic research addressing many of the required criteria and develop new tools and systems to lead the field in exciting new interdisciplinary directions and control strategies. The appearance of the avian-like H3N2 CIV in the US in 2015 presented an opportunity to test the CEIRS network pre-pandemic risk assessment plan pipeline. Upon selection of the viral strains, groups were assigned specific tasks and time to completion was monitored. Available viral isolates were distributed and generation of reverse genetics derived viruses from publicly available sequences was initiated within weeks. In 2 months, the network demonstrated that the H3 CIV preferentially bound to α2,3 sialic acid receptors but replicated efficiently in primary human nasal and bronchial epithelial cells; replicated and caused disease in DBA2/J mice and transmitted effectively in direct contact ferrets and guinea pigs without causing clinical signs of infection. Using sera from human cohorts located in the US, Asia and South America, we demonstrated that younger subjects born after 1970 had no cross-protective humoral immunity, even if vaccinated for seasonal human H3 viruses, and may be considered at high-risk for H3 CIV infections. There was, however, a high rate of pre-existing NA cross-reactive Abs to H3N2 CIV, especially in the older population. Given the evidence that NA antibodies are associated with decreased disease and shortened duration of viral shedding [53,54], it is possible that the NA antibodies would reduce pathogenesis in and transmission. Importantly, if an H3 CIV did emerge in humans, our studies suggest the available FDA-approved antiviral drugs could be used and hmAbs could be effective for passive immunization.

The CEIRS network study goes beyond characterization of emerging viruses. It highlights new approaches (e.g. AIR), tools (e.g. primary human respiratory cells and explants; reporter viruses) and involved hypothesis-driven work (i.e. identifying cross-reactive hmAbs) that were only possible due to ongoing fundamental basic research. This was the first study, to our knowledge, to directly compare transmission in the ferret and guinea pig models. Ongoing studies within the CEIRS network are expanding this comparison to other viral strains. We also provided the initial comparison of viral replication in isolated primary respiratory bronchial epithelial cells and bronchus explants demonstrating that the results differ. In contrast to the human viruses, the CIVs replicated, although not to high titers, in the hBECs but failed to replicate. This could be due to donor variability, differences in sialic acid linkage distribution, or the impact of having a mixed culture of lung cells in the explant. Ongoing studies are aimed

at defining these differences and identifying the best system for characterization of emerging viruses; especially avian and mammalian strains. Overall, the outcome of the current and ongoing studies will provide important new information to fundamental research, but also to refine and update the public health algorithms.

Several critical lessons were learned during this exercise. First–having a sequence does not guarantee that you can generate a virus. We specifically included H3 CIV strains that would need to be synthesized and rescued by reverse genetics as this is the most likely scenario [55,56]. Attempts to rescue complete H3 recombinant canine viruses were unsuccessful. This likely resulted from the lack of 5' and 3' untranslated region (UTR) sequence information in the public databases. UTRs are critical for gene expression [57]. However, 6+2 viruses containing CIV HA and NA on a PR8 backbone were created and distributed to all groups leading to the second lesson learned. Several groups were unable to receive WT and/or recombinant viruses due to technology transfer issues, and material transfer agreements (MTA) and patent rights between institutions were barriers to sharing of these reagents. While outside the control of the CEIRS network investigators, this had a major impact on the exercise, the timeliness of the studies and was the reason the viruses tested in experiments differed. An outcome of the exercise was establishment of universal MTAs between all CEIRS institutions and agreements that in the case of a public health emergency, public health need would overshadow commercial considerations and patent rights. While not an issue for this exercise, previous situations have shown that obtaining appropriate import permits and biosafety approvals from various government agencies (USDA, CDC, etc.) can lead to significant delays in initiating research efforts. This is especially true with emerging viruses from non-human hosts where different government agencies may need time to determine which agency holds jurisdiction and what permits and/or biosafety conditions will be required. Although we are confident that the CEIRS network could rapidly respond to an emerging virus or pandemic (as we did in during the 2009 pandemic and in response to the H7N9 virus in 2013) by redirecting existing resources, including personnel, to provide critical fundamental research on novel influenza strains our response time can always be improved. Our pipeline, coupled with the global surveillance efforts in humans and other mammals, birds and at the animal-human interface, may identify the emerging viruses of concern.

Collectively, this collaborative CEIRS exercise demonstrated our ability to respond to a potentially emergent virus with putative risk to humans, generating reagents and assays critical to assessing risk of infection and pathogenesis to humans. The lessons learned from this exercise go beyond influenza virus, being important for studies of emerging infectious diseases of all types.

## Materials and methods

### Ethics statement

All experiments were approved and conducted in appropriate animal biosafety level facilities in compliance with the policies of the National Institutes of Health, the United States Animal Welfare Act, in strict compliance with European guidelines (European Union [EU] directive on animal testing 86/609/EEC) and Dutch legislation (Experiments on Animals Act, 1997) and with the approval of the University of Georgia (#A2017 05–009), St Jude Children's Research Hospital (535 and 428), Emory University (PROTO201700595), Icahn School of Medicine at Mount Sinai (IACUC-2013-1408), University of Rochester (101890 / 2014-019E) and Erasmus University (Erasmus MC permit AVD101002015340, protocol number 15-340-07) Animal Care and Use Committees and Institutional Biosafety Committees. Animals were euthanized following AVMA approved guidelines. All human samples were collected upon written and/or

oral consent on studies approved by the DMID, and the University of Rochester (NCT03328325), Johns Hopkins (IRB00091667), St Jude Children's Research Hospital (INDI-FLU) and University of Hong Kong (UW 14–119) individual institutional review boards. All data analyzed were anonymized.

## Viruses

A complete listing of viruses and abbreviations can be found in Table 2. HK68 and WI05 were obtained from BEI Resources (NR-28620 and NR-41800, respectively), propagated in Madin-Darby canine kidney (MDCK) cells and p1 stocks used in experiments. The CIV-41915 virus was originally isolated in MDCK cells by the Cornell Animal Health and Diagnostic Center and MDCK p2 stocks were distributed to the network from Dr. Parrish's laboratory. The Beth15 H3N2 human strain was isolated in human nasal epithelial cell cultures and further propagated in MDCK cells. Vials of p2 working stock was distributed to the CEIRS network from Dr. Pekosz laboratory. All viruses were used at p1 or p2 stocks or as stated in the specific methods.

## Reverse genetics

Recombinant viruses were generated using plasmid-based reverse genetics techniques as previously described [58–60].

## Cells

MDCK (ATCC CCL-34 and IRR FR-58) and Vero (ATCC CCL-81) cells were grown in Dulbecco's modified Eagle's medium (DMEM) supplemented with 5–10% fetal bovine serum (FBS), 2 mM GlutaMax, 100 U/ml Penicillin and 100 μg/ml Streptomycin. Human bronchial epithelial cells (hBECs) were obtained from Lonza (CC-2540) and grown in expansion media (Airway Epithelial Cell Growth Medium, Promocell) prior to seeding on transwell inserts. Cells were maintained at an air-liquid interface (ALI) for three weeks to allow differentiation. Human nasal epithelial cells (hNECs) were obtained from disease-free tissues and maintained at ALI as previously described [22].

## Ex-vivo culture of human respiratory tract explants

Fresh residual tissues from normal human bronchus and lung were obtained from patients undergoing surgical resection at Queen Mary Hospital, Hong Kong and cells cultured as previously described [61–63]. Briefly, bronchial tissue fragments were placed on a surgical sponge (Simport) floating in a 24-well tissue culture plate containing F-12K nutrient mixture (Gibco), 100 U/ml penicillin, and 100 μg/ml streptomycin, to create an air-liquid interface. Lung explants were cultured similarly but without the use of surgical sponge. All tissues were maintained at 37°C, 5% $CO_2$.

## Multicycle growth kinetics

**MDCK cells.** Cells were plated at $5 \times 10^5$ cells/well in triplicate in 12-well tissue culture plates, grown to confluency and infected at a MOI of 0.001 [64]. After 1 hour of adsorption at room temperature, fresh media supplemented with 1 μg/ml TPCK-treated trypsin (Sigma) was added and cells were placed at 33°C, 37°C or 39°C. Tissue culture supernatants were collected at the designated times post-infection and viral titers were determined by immunofocus assay (fluorescent forming units, FFU/ml) using mouse mAbs against NP including: HB-65 (ATCC,

H16-L10-4R5) for rCIV-23 H3N8 and rWY03 H3N2; and HT-103 for H3N2 rCIV-11613 [65]. Mean values and standard deviations (SDs) were calculated using Microsoft Excel software.

**hNECs and hBECs.** Differentiated cells at ALI were equilibrated for 2 days at 32˚C or 37˚C, washed with 150 μl of infection media (DMEM supplemented with 0.3% BSA [Sigma], 100 U/mL penicillin, 100 μg/mL streptomycin, 2 mM GlutaMAX [Gibco]) and the basolateral media changed immediately before infection with MOI of 0.1. Virus was incubated for 2 hours at 32˚C, washed and cells incubated at 32˚C or 37˚C. At the indicated times, 100 μl of media was added to the apical chamber incubated for 10 minutes at 32˚C, collected and viral titers determined by tissue culture infectious dose 50 ($TCID_{50}$) assay [66]. Endpoint calculations were determined by the Reed-Muench algorithm [67].

***Ex-vivo* human respiratory tract explants.** Human respiratory tract explant cultures were submerged in 1 ml of virus at ~$10^6$ $TCID_{50}$/ml for 1 hour at 37˚C, carefully washed with PBS to remove unbound virus, and placed onto a surgical sponge in a 24-well tissue culture plate filled with 1 ml/well of culture medium containing F-12K nutrient mixture (Gibco), 100 U/ml penicillin, and 100 μg/ml streptomycin, to create an ALI and maintained at 37˚C. Supernatant from culture media of virus infected bronchial tissues was collected at 1, 24, and 48-hour post-infection (hpi) and titrated by $TCID_{50}$ assay on MDCK cells. Experiments were performed with tissues from 3 donors (n = 3).

## Plaque assays

Plaque assays were performed on MDCK cells plated at $1x10^6$ cells/well in 6-well tissue culture plates and grown to confluency as described [68]. $TCID_{50}$ assays were performed on confluent MDCK cells in 96-well tissue culture plates as described [68]. Endpoint calculations were determined by the Reed-Muench algorithm [67].

## HA acid stability

HA acid stability was measured by syncytia assay as described [69]. Briefly, 17 to 24 hpi, HA-expressing Vero cells were incubated for 5 minutes with TPCK-treated trypsin and pH-adjusted PBS buffers, the pH then neutralized by washing with PBS, and then incubated in regular medium for 3 hours at 37˚C. Subsequently, the cells were fixed and stained for microscopy.

## Acid inactivation and luciferase infectivity assay

To measure the abilities of the viruses to retain or lose infectivity as a function of exposure to solutions of varying acid pH, virus stocks were incubated 1 hour at 37˚C in pH-adjusted PBS solutions before the pH was neutralized. Viruses were subsequently used to infect MDCK Luc9.1 reporter cell (generous gift of Dr. Ruben Donis). *Renilla* luciferase enzymatic activity was used to assay for retention of infectivity. The susceptibility to inactivation by exposure to low pH was reported as $pH_{50}$.

## Fluorescent labeling of viruses and binding to CFG glycan microarray

To fluorescently label viral particles, 200 μl of virus was incubated with 25 μg of Alexafluor 488 (Invitrogen) in 1 M $NaHCO_3$ (pH 9.0) for 1 hour at room temperature. Excess label was removed via dialysis in PBS using a 7,000-molecular-weight-cutoff Slide-A-Lyzer mini-dialysis unit (Thermo Scientific) overnight at 4˚C. In all cases, labeled viruses were used in experiments the following day. Labeled virus was incubated on the glycan microarray (CFG version 5.3 from the National Center for Functional Glycomics) for 1 hour at 4˚C, to inhibit viral NA

activity, then the slide was washed to remove unbound virus and scanned using a ProScanArray microarray scanner (Perkin Elmer) at 495 nm (ex) and 519 nm (em) for Alexa Fluor 488 fluorescence.

## Immunofluorescence microscopy

MDCK cells were plated at $1 \times 10^5$ cells/well in 48-well tissue culture plates and grown to confluency. Cells were then mock or infected with the indicated viruses at a MOI of 3 and incubated at 33˚C. At 12 hpi, cells were fixed with 4% paraformaldehyde and permeabilized for 15 minutes at room temperature with 0.5%Triton X-100. After incubating in block solution (2.5% BSA in PBS) for 1 hour at room temperature, cells were incubated with the following mAbs (1μg/ml) or pAbs (dilution 1:1000) diluted in blocking solution for 1 hour at 37˚C: anti-HA A/duck/Shantou/1283/2001 H3N8 goat pAb (BEI Resources NR-34586), A/equine/Miami/1/1963 H3N8 mouse mAb equine 7.1, or A/Hiroshima/52/2005 mouse pAb (BEI Resources F-287) for the staining of rCIV H3N2-, rCIV H3N8- or r/Wy H3N2-infected cells, respectively; anti-NA A/Singapore/1/1957 H2N2 goat pAb (BEI Resources NR-3137), A/equine/Miami/1/1963 H3N8 goat pAb (BEI Resources NR-3145) or hmAb 109219 (kindly provided by James J. Kobie) for the staining of rCIV H3N2-, rCIV H3N8- or r/Wy H3N2-infected cells, respectively; anti-NS1 1A7and 4D4 [70] mouse mAbs cocktail (1:1, 2 μg/ml). After incubation with the primary antibodies, cells were washed and incubated with a 1:200 dilution of fluorescein isothiocyanate (FITC)-conjugated anti-mouse (Dako), anti-goat (Jackson Immuno Research) or anti-human (Jackson Immuno Research) secondary antibodies in blocking solution, along with 4′,6-diamidino-2-phenylindole (DAPI; Research Organics) for 1 hour at 37˚C. After washing, fluorescent staining and mCherry expression were evaluated and images were captured using a fluorescent microscope (Olympus IX81 with camera QIMAGING, Retiga 2000R).

## Animal experiments

**Mice.**    Six- to eight-week-old C57BL/6 (National Cancer Institute, NCI), BALB/c (Envigo) or DBA/2J (The Jackson Laboratories) were lightly anesthetized with isoflurane and intranasally (i.n.) inoculated with 50 μl of $10^6$ TCID$_{50}$ of CIV-41915 p3, rCIV-1177 p3, Beth55 p4 or DMEM diluent. Mice were monitored daily for clinical disease signs and weighed every 48 hours. Viral titers in whole lungs were determined at 3 and 5 dpi (n = 5 mice per group) by TCID$_{50}$. Seroconversion was determined at 26 dpi.

**Ferrets.**    Influenza virus seronegative 4 to 6-month-old outbred male ferrets (Triple F Farms, PA, n = 9 per virus and n = 3 per group) were lightly anesthetized and intranasally inoculated with $10^6$ TCID$_{50}$ of virus (p3) in 1.0 ml sterile phosphate-buffered saline. At 1 dpi, one naïve contact ferret was placed in the same cage with a directly inoculated ferret and one aerosol contact was placed in an adjacent cage separated by a wire grill. All ferrets were monitored daily for weight loss, body temperature, and clinical signs of infection for 12 dpi [71] [72]. To monitor virus shedding, nasal washes were collected every other day for 12 dpi and viral titers determined by TCID$_{50}$. Seroconversion was determined at 21 dpi.

**Guinea pigs.**    CIV-41915 was passaged once in embryonated hen eggs and contact transmission experiments were performed as described [73]. Briefly, female Hartley strain guinea pigs (Charles River Laboratories) were intranasally inoculated with 10, 100 and 1,000 PFU (n = 4 per group). At 24 hpi, each inoculated animal was co-housed with a naïve contact animal for seven days. Nasal washes were collected from anesthetized animals on days 2, 4, 6 and 8 post-inoculation, and stored at -80˚C prior to viral quantitation by plaque assay on MDCK cells. Experiments were performed under conditions of 10˚C and 20% relative humidity and repeated three times.

## Susceptibility to antiviral drugs

Stocks of oseltamivir carboxylate, zanamivir, peramivir, and amantadine hydrochloride (amantadine) (Sigma) were prepared in distilled water, filter-sterilized, and stored in aliquots at –20ºC. Susceptibility to NA inhibitors with WT viruses was assessed in a fluorescence-based assay using 100 µM fluorogenic substrate 2'-(4-methylumbelliferyl)-α-D-N-acetylneuraminic acid (MUNANA) (Sigma) [74]. $IC_{50}$ values were calculated using GraphPad Prism 6 software (GraphPad Software, La Jolla, CA). Phenotypic susceptibility to amantadine was assessed using biological assays in MDCK cells [28].

## Sequence-based maps

Viral protein sequences and associated metadata for H3, N2, and N8 influenza virus subtypes were obtained from the Influenza Research Database (fludb.org). For quality control, sequences that were either missing or contained erroneous amino acids were removed. Sequences for each viral protein were aligned using MUSCLE algorithm [75]. Hamming distances were calculated pairwise for all aligned protein sequences within each group (i.e. H3, N2, N8 stains) using the *stringdist* package in R [76]. Dimensional reduction of the resulting Hamming distance matrix was performed using *cmdscale* function in the base package of R. Goodness-of-fit (GOF) calculations were performed as previously described [32].

## Enzyme-linked immunosorbent assay (ELISA)

Canine and human influenza virus-specific IgG levels in human sera were measured by ELISA. Briefly, MaxiSorp 96-well plates (Nunc) were coated with optimized dilutions of CIV-41915, rCIV-23 H3N8 or WY03 H3N2 infected or mock-infected MDCK cell extracts. Serial 3-fold sera dilutions (starting at 1/20) were added to blocked (1% BSA in PBS) plates, followed by a 2-hour incubation at room temperature. Bound antibodies were detected by addition of alkaline phosphatase-conjugated anti-human IgG (Mabtech cloneMT78), followed by *p*-nitrophenyl phosphate substrate. After color development, the reaction was stopped by addition of 0.5 M NaOH and absorbance was read at 405 nm. IgG antibody titers were determined by comparison of parallel sample titrations in wells coated with infected or mock-infected cell extracts. The titer was defined as the reciprocal of the highest sample dilution that gave an OD 2-fold greater against the infected versus the mock-infected cell extracts. Titers < 20 were assigned an arbitrary value of 10 for analysis.

## HAI assays

Sera samples were treated with receptor-destroying enzyme (RDE, Denka Seiken) overnight at 37˚C, heat inactivated for 30 minutes at 56˚C and HAI assays performed with turkey red blood cells (RBCs) as described [77]. To evaluate the neutralizing activity of hmAbs, HAI assays were performed as described above with a starting concentration of 0.01 mg/ml of each of the hmAbs.

## Serum neutralization assays

Two-fold serial dilutions of RDE-treated sera in virus infection media (DMEM containing penicillin, streptomycin, 0.5% BSA, and 5 µg/mL TPCK-treated trypsin) were mixed with 100 $TCID_{50}$ of each virus for 1 hour at room temperature. Confluent MDCK cells were inoculated with the sera and virus mixture in quadruplicate wells for 24 hours at 32˚C. Following the 24-hour incubation, the inoculum was removed, the cells were washed, new infection media added and incubated at 32˚C until CPE was observed. The cells were then fixed with 4%

formaldehyde and stained with Napthol blue black at room temperature overnight. The neutralizing antibody titer was calculated as the highest serum dilution that eliminated virus cytopathic effects in 50% of the wells.

### Fluorescent-based microneutralization assay (FMNA)

To assess the neutralization activity of hmAbs a FMNA was performed as previously described with slight modifications [42]. MDCK cells were plated at $5x10^4$ cells/well in 96-well tissue culture plates, incubated overnight and infected with the mCherry-expressing rCIV or WY03 H3N2 at a MOI of 0.005 in triplicate and incubated with media supplemented with 2-fold serial dilutions of hmAbs (starting concentration of 0.02 mg/ml). At 60 hpi, cells were washed with PBS and fluorescence was evaluated by fluorescence microscopy and quantified using a fluorescence plate reader (SpectralMax iD3, Molecular Devices) [42,43]. Infected cells in absence of hmAb represented 100% of infection. Cells in absence of viral infection were used to calculate the fluorescence background. Sigmoidal dose-response curves were generated (GraphPad Prism 7.0c) to calculate $NT_{50}$values.

### Enzyme-linked lectin assay (ELLA)

The inhibition of NA enzymatic activity by human sera samples was evaluated by ELLA as previously described [78,79]. Briefly, recombinant PR8 viruses expressing H4 HA (A/Mallard/Netherlands/1/1999 H4N6, BEI Resources NR-28996) and the NA from rCIV-11613, rCIV-23, or WY03 H3N2 viruses were used. Microtiter 96-well plates were coated with 100 μl of 50 μg/ml fetuin (Sigma) and incubated at 4˚C overnight. The following day, heat-inactivated (30 minute at 56˚C) human sera samples were 2-fold serially diluted (starting dilution 1/16) and mixed with the virus. The dilutions were then incubated on the fetuin-coated plates for 18 hour at 37˚C and bound fetuin detected by incubation with 100 μl of peanut agglutinin conjugated to horseradish peroxidase (PNA-HRP) (1 μg/ml, Sigma) for 2 hour at room temperature followed by incubation with 3,3',5,5' tetramethylbenzidine substrate (TMB, Sigma) for 15 minutes at room temperature. Optical density was measured at 450 nm. For each assay, wells containing only virus (maximum signal) and wells containing only PBS-1% BSA (background signal) were included as internal controls. For analysis, background absorbance values were subtracted from the absorbance obtained from the human sera. The sera dilution at which 50% of the maximum signal was inhibited was concluded to be the $NAI_{50}$ titer. When NAI titers were below the limit of detection ($< 16$), an arbitrary value of 8 was assigned.

### Arrayed imaging reflectometry (AIR)

Preparation of hmAbs and label-free microarray sensor was described in detail elsewhere [44,80]. For high-multiplex sensing, duplicate spots were printed for each hmAb. Anti-FITC antibody spots were printed adjacent to each hmAb probe and used as the baseline for detection. Human IgG was spotted as a negative control for non-specific binding. Purified bovine IgG secondary antibodies were used for quality control of array printing. Experiments with human H3N2 vaccine strains and H3N2 and H3N8 rCIVs were conducted in 96-well plates. Two blocking solutions consisting of 10 mg/ml BSA in sodium acetate buffer (50 mM at pH 5.0) and 10% FBS in modified PBS-EDTA-Tween 20 (10 mM PBS, 5 mM EDTA, and 0.5% Tween 20 at pH 7.4) assay wash buffer (AWB) were prepared. After blocking, the chips were transferred for virus exposure. Solutions of human H3N2 strains and H3N2 and H3N8 rCIVs were prepared in 10-fold dilutions in 10% FBS in AWB. Blank 10% FBS solutions were used as negative control groups. In each case, three chips per condition were incubated overnight at room temperature, then washed in AWB several times. Finally, the chips were rinsed in

distilled water and dried in nitrogen before array imaging. Once dried, chips were imaged immediately on a prototype AIR reader (Adarza BioSystems, Inc). AIR images were acquired in a 16-bit TIFF format, with exposure times varied from 50 ms to 1 second. The AIR image files were then analyzed using NIH-ImageJ (version 1.46r). The median value of the reflection intensities averaged from three chips represents the amount of material bound to each hmAb, and these data were normalized relative to the positive control (human IgG). All experiments were repeated at least twice.

## Cross-reactivity of hmAbs by immunofluorescence and HeatMap

MDCK cells were plated at $1 \times 10^5$ cells/well in a 48-well tissue culture plate and mock-infected or infected with rCIV-41915, WY03 or HK68 H3N2 at a MOI of 3 at 33˚C in triplicate. At 12 hpi, cells were fixed with 4% paraformaldehyde (PFA) and permeabilized for 15 minutes at room temperature with 0.5%Triton X-100 in PBS. Cells were incubated with 1 µg/ml of the indicated anti-HA hmAbs diluted in 2.5% BSA in PBS blocking solution for 1 hour at 37˚C followed by an incubation with a 1:200 FITC-conjugated anti-human secondary Ab (Jackson Immuno Research). Fluorescence was evaluated under a fluorescent microscope (Olympus IX81) and images were captured (QIMAGING, Retiga 2000R). The fluorescence intensity of these images was measured using the ImageJ 1.51s software and data was displayed using a Heatmap visualization method (GraphPad Prism 7.0c). For analysis, the hmAb displaying the maximum signal for each viral HA was considered 100% and used to normalize data to reflect relative binding of each hmAb.

## Statistical analysis

Statistical analysis was performed using the statistical package SPSS version 20.0 or Prism (GraphPad, La Jolla, CA). *In vivo* and *in vitro* studies were analyzed using two-tailed Student's test. Serum IgG titers are expressed at geometric mean±95% CI. Differences between medians of pre-existing human antibodies were compared using Mann-Whitney U test. Correlation between HAI and MN titers were analyzed by Pearson correlation test. P-values <0.05 were considered significant.

## Supporting information

**S1 Fig. Cross-reactivity of hmAbs by immunofluorescence.** MDCK cells were infected (MOI of 3) with H3N2 or H3N8 rCIV, rWy03 H3N2 and HK68 H3N2. At 12 hpi cells were fixed, permeabilized and incubated with 1 µg/ml of the indicated hmAbs. After incubation with a secondary anti-human FITC-conjugated Ab, fluorescence was imaged under a fluorescent microscope. The hmAbs were grouped based on their ability to recognize H3N2 and H3N8 rCIV, and rWy03 H3N2. Representative images of the reactivity of the hmAbs against infected cells are shown in the right. The scale represents % of recognition. Scale bars, 200 $\mu$m.
(TIF)

**S1 Table. Human HAI titers against H3 CIV.** Sera was tested for HAI titers against select H3 CIVs.
(DOCX)

## Author Contributions

**Conceptualization:** Luis Martinez-Sobrido, Hanyuan Zhang, Jeanne Holden-Wiltse, Sanjukta Bandyopadhyay, Brian R. Wasik, Benjamin L. Miller, Patrick C. Wilson, Mark Y. Sangster,

David J. Topham, David A. Steinhauer, Richard D. Cummings, Jasmina M. Luczo, Stephen M. Tompkins, John Steel, Anice C. Lowen, Sabra L. Klein, Farah el Najjar, Andrew Pekosz, Colin R. Parrish, Ian E. H. Voorhees, Yoshihiro Kawaoka, Gabriele Neumann, Shufang Fan, Masato Hatta, Huihui Kong, Gongxun Zhong, Guojun Wang, Adolfo García-Sastre, Sander Herfst, Ron Fouchier, David Burke, David Pattinson, Derek J. Smith, Victoria Meliopoulos, Sean Cherry, Charles J. Russell, Scott Krauss, Angela Danner, Malik Peiris, R. A. P. M. Perera, M. C. W. Chan, Elena A. Govorkova, Gavin Smith, Yao-Tsun Li, Paul G. Thomas, Stacey Schultz-Cherry.

**Data curation:** Luis Martinez-Sobrido, Pilar Blanco-Lobo, Laura Rodriguez, Sanjukta Bandyopadhyay, David J. Topham, Lauren Byrd-Leotis, David A. Steinhauer, Richard D. Cummings, John Steel, Anice C. Lowen, Hui Tao, Farah el Najjar, Kathryn Shaw-Saliba, Ian E. H. Voorhees, Yoshihiro Kawaoka, Gabriele Neumann, Masato Hatta, Guojun Wang, Sander Herfst, David Burke, Derek J. Smith, Sean Cherry, Elena A. Govorkova, Gavin Smith, Stacey Schultz-Cherry.

**Formal analysis:** Luis Martinez-Sobrido, Pilar Blanco-Lobo, Laura Rodriguez, Theresa Fitzgerald, Hanyuan Zhang, Phuong Nguyen, Christopher S. Anderson, Jeanne Holden-Wiltse, Sanjukta Bandyopadhyay, Aitor Nogales, Marta L. DeDiego, Brian R. Wasik, Benjamin L. Miller, Carole Henry, Patrick C. Wilson, Mark Y. Sangster, John J. Treanor, David J. Topham, Lauren Byrd-Leotis, David A. Steinhauer, Richard D. Cummings, Jasmina M. Luczo, Stephen M. Tompkins, Cheryl A. Jones, John Steel, Anice C. Lowen, Ashley L. Fink, Sabra L. Klein, Nicholas Wohlgemuth, Farah el Najjar, Andrew Pekosz, Colin R. Parrish, Ian E. H. Voorhees, Yoshihiro Kawaoka, Gabriele Neumann, Shiho Chiba, Shufang Fan, Masato Hatta, Huihui Kong, Gongxun Zhong, Lucas M. Ferreri, Sander Herfst, Mathilde Richard, David Pattinson, Derek J. Smith, Victoria Meliopoulos, Pamela Freiden, Bridgett Sharp, Charles J. Russell, Richard J. Webby, Scott Krauss, Angela Danner, Malik Peiris, R. A. P. M. Perera, M. C. W. Chan, Elena A. Govorkova, Bindumadhav M. Marathe, Philippe N. Q. Pascua, Gavin Smith, Yao-Tsun Li, Paul G. Thomas, Stacey Schultz-Cherry.

**Funding acquisition:** Luis Martinez-Sobrido, Patrick C. Wilson, John J. Treanor, David J. Topham, David A. Steinhauer, Richard D. Cummings, Stephen M. Tompkins, John Steel, Anice C. Lowen, Sabra L. Klein, Andrew Pekosz, Richard E. Rothman, Yoshihiro Kawaoka, Adolfo García-Sastre, Daniel R. Perez, Ron Fouchier, Derek J. Smith, Charles J. Russell, Richard J. Webby, Malik Peiris, Elena A. Govorkova, Paul G. Thomas, Stacey Schultz-Cherry.

**Investigation:** Luis Martinez-Sobrido, Pilar Blanco-Lobo, Richard D. Cummings, Kaori Sakamoto, Ashley L. Fink, Sabra L. Klein, Farah el Najjar, Andrew Pekosz, Colin R. Parrish, Ian E. H. Voorhees, Gabriele Neumann, Shufang Fan, Masato Hatta, Huihui Kong, Gongxun Zhong, Guojun Wang, Lucas M. Ferreri, Mathilde Richard, David Burke, Victoria Meliopoulos, Brandi Livingston, Charles J. Russell, Scott Krauss, Malik Peiris, Elena A. Govorkova, Stacey Schultz-Cherry.

**Methodology:** Luis Martinez-Sobrido, Pilar Blanco-Lobo, Laura Rodriguez, Theresa Fitzgerald, Phuong Nguyen, Aitor Nogales, Marta L. DeDiego, Brian R. Wasik, Benjamin L. Miller, Carole Henry, David J. Topham, Lauren Byrd-Leotis, David A. Steinhauer, Jasmina M. Luczo, Kaori Sakamoto, Cheryl A. Jones, John Steel, Shamika Danzy, Hui Tao, Ashley L. Fink, Nicholas Wohlgemuth, Katherine J. Fenstermacher, Farah el Najjar, Colin R. Parrish, Yoshihiro Kawaoka, Gabriele Neumann, Shiho Chiba, Shufang Fan, Masato Hatta, Huihui Kong, Gongxun Zhong, Guojun Wang, Daniel R. Perez, Lucas M. Ferreri, Sander Herfst, Mathilde Richard, Victoria Meliopoulos, Pamela Freiden, Brandi Livingston, Bridgett

Sharp, Sean Cherry, Guohua Yang, Charles J. Russell, Subrata Barman, Richard J. Webby, Scott Krauss, Angela Danner, Karlie Woodard, Malik Peiris, R. A. P. M. Perera, M. C. W. Chan, Elena A. Govorkova, Bindumadhav M. Marathe, Philippe N. Q. Pascua, Paul G. Thomas, Stacey Schultz-Cherry.

**Project administration:** Luis Martinez-Sobrido, Andrew Pekosz, Lauren Sauer, Mitra K. Lewis, Kathryn Shaw-Saliba, Richard E. Rothman, Zhen-Ying Liu, Kuan-Fu Chen, Yoshi-hiro Kawaoka, Gabriele Neumann, Melissa B. Uccellini, Adolfo García-Sastre, Victoria Meliopoulos, Juan Carlos Dib, Scott Krauss, Gavin Smith, Stacey Schultz-Cherry.

**Resources:** Luis Martinez-Sobrido, Christopher S. Anderson, Aitor Nogales, Marta L. DeDiego, Brian R. Wasik, Patrick C. Wilson, Lauren Sauer, Yoshihiro Kawaoka, Gabriele Neumann, Shiho Chiba, Shufang Fan, Juan Carlos Dib, Scott Krauss, Malik Peiris, Stacey Schultz-Cherry.

**Supervision:** Luis Martinez-Sobrido, Benjamin L. Miller, John J. Treanor, David J. Topham, David A. Steinhauer, Richard D. Cummings, Stephen M. Tompkins, John Steel, Anice C. Lowen, Sabra L. Klein, Andrew Pekosz, Richard E. Rothman, Zhen-Ying Liu, Kuan-Fu Chen, Colin R. Parrish, Yoshihiro Kawaoka, Adolfo García-Sastre, Daniel R. Perez, Sander Herfst, Ron Fouchier, Derek J. Smith, Victoria Meliopoulos, Juan Carlos Dib, Charles J. Russell, Richard J. Webby, Scott Krauss, M. C. W. Chan, Elena A. Govorkova, Gavin Smith, Paul G. Thomas, Stacey Schultz-Cherry.

**Validation:** Luis Martinez-Sobrido, Yoshihiro Kawaoka, Derek J. Smith, Stacey Schultz-Cherry.

**Visualization:** Jeanne Holden-Wiltse, Kaori Sakamoto, Farah el Najjar, Gavin Smith, Stacey Schultz-Cherry.

**Writing – original draft:** Luis Martinez-Sobrido, Hanyuan Zhang, Benjamin L. Miller, David J. Topham, Ashley L. Fink, Sabra L. Klein, Farah el Najjar, Andrew Pekosz, Victoria Meliopoulos, Charles J. Russell, Malik Peiris, M. C. W. Chan, Elena A. Govorkova, Stacey Schultz-Cherry.

**Writing – review & editing:** Luis Martinez-Sobrido, Brian R. Wasik, Benjamin L. Miller, Carole Henry, Patrick C. Wilson, John J. Treanor, David J. Topham, David A. Steinhauer, Richard D. Cummings, Jasmina M. Luczo, Stephen M. Tompkins, John Steel, Anice C. Lowen, Sabra L. Klein, Farah el Najjar, Andrew Pekosz, Colin R. Parrish, Ian E. H. Voorhees, Yoshihiro Kawaoka, Gabriele Neumann, Daniel R. Perez, Ron Fouchier, Derek J. Smith, Victoria Meliopoulos, Charles J. Russell, Scott Krauss, Malik Peiris, M. C. W. Chan, Elena A. Govorkova, Gavin Smith, Paul G. Thomas, Stacey Schultz-Cherry.

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
