## [Decision Letter · Decision Letter 0]

4 Nov 2019

Dear Dr. Schultz-Cherry:

Thank you very much for submitting your manuscript "Characterizing Emerging Canine H3 Influenza Viruses: Challenging the Centers of Excellence for Influenza Research and Surveillance (CEIRS) Network Pipeline." (PPATHOGENS-D-19-01711) for review by PLOS Pathogens. Your manuscript was fully evaluated at the editorial level and by independent peer reviewers. The reviewers appreciated the attention to an important topic but identified some aspects of the manuscript that should be improved.

As you know, this manuscript is quite unusual, and as you can see particularly in comments from reviewers 1,2 there're some comments addressing the proposition of the manuscript that should be carefully addressed and clarified. In addition there're a number of comments concerning the scientific merits that, given the large number of collaborating laboratories, can be addressed in the context of a "minor" revision (though for one individual lab it would probably be a major revision). Finally, please check the entire manuscript for inconsistencies (methods, figure panels, writing style ect). As suggested by reviewer 2 it might be best if a single authors goes through the whole manuscript. 

We therefore ask you to modify the manuscript according to the review recommendations before we can consider your manuscript for acceptance. **Your revisions should address all the specific points made by each reviewer. Please make sure that the wording of the manuscript and its title do not sound as a "sales pitch" and do a careful proofreading and editing of the final manuscripts.** 

(1) A letter containing a detailed list of your responses to the review comments and a description of the changes you have made in the manuscript. Please note while forming your response, if your article is accepted, you may have the opportunity to make the peer review history publicly available. The record will include editor decision letters (with reviews) and your responses to reviewer comments. If eligible, we will contact you to opt in or out.

(2) Two versions of the manuscript: one with either highlights or tracked changes denoting where the text has been changed; the other a clean version (uploaded as the manuscript file).

We hope to receive your revised manuscript within 60 days or less. If you anticipate any delay in its return, we ask that you let us know the expected resubmission date by replying to this email.

[LINK]

Sincerely,

Carolina B. Lopez, Ph.D.

Associate Editor

PLOS Pathogens

Volker Thiel

Section Editor

PLOS Pathogens

Kasturi Haldar

Editor-in-Chief

PLOS Pathogens

orcid.org/0000-0001-5065-158X

Grant McFadden

Editor-in-Chief

PLOS Pathogens

orcid.org/0000-0002-2556-3526

Reviewer's Responses to Questions

**Part I - Summary**

Reviewer #1: This paper describes the efforts of a large consortium of CEIRS-funded investigators to perform quantitative risk assessment on the potential for human emergence of recently circulating canine H3 viruses. This is meant as a proof-of-principle of the utility of the CEIRS network for supporting large, multi-investigator efforts aimed at facing specific public health threats posed by influenza viruses. They describe a comprehensive assessment of phenotypic correlates of emergence potential for a small sample of canine H3N2 and H3N8 viruses. They show variable ability of canine viruses to replicate in primary human cells from upper vs. lower airway, that they can readily transmit via contact between guinea pigs, receptor binding profiles and pH characteristics of HA, that the viruses are sensitive to licensed antivirals, and that people born after 1970 have limited immune recognition of canine H3s. Plus a bunch of other stuff that I don’t need to list. Altogether, they conclude that canine H3 viruses pose a very limited emergence threat and suggest that they would be effectively controlled by existing antivirals and broadly-reactive anti-H3 therapeutic monoclonal Abs.

Overall, the paper presents a nice, comprehensive data set that addresses an important open question. With a couple of minor exceptions, the conclusions are well supported by the data, and the experimental approaches used are sound. Given the huge number of labs involved, the paper is well-written and cohesive, but is a little sloppy with some missing or unexplained figures and poor descriptions of some of the data/rationales. It also comes across as weirdly sales-pitchy in terms of the CEIRS angle. The paper would benefit from toning that down a bit and focusing on the science.

Reviewer #2: This is an unusual manuscript to be asked to review since its prime claim seems to be whether the NIAID CEIRS collaboration was able to provide research results that can be used to perform a risk assessment of the pandemic potential of influenza viruses from dogs. The risk assessment frameworks for the results to be applied to are the Influenza Risk Assessment Tool (IRAT), developed by CDC, and the Tool for Influenza Pandemic Risk Assessment (TIPRA), developed by WHO. The unusual question that has been posed is reflected in the unusual type of title of the manuscript.

Reviewer #3: In the research article entitled “Characterizing Emerging Canine H3 Influenza Viruses: Challenging the Centers of Excellence for Influenza Research and Surveillance (CEIRS) Network Pipeline”, the authors test the efficiency of the CEIRS network to respond to emerging threats. It is critical to perform these exercises to ensure productive cross-network interactions and assess response times. Importantly, the authors identify key areas, based on this exercise, that can be improved upon for future studies with emerging pandemic threats. This assessment is not only useful for the CEIRS network but also for other scientific centers tasked with addressing large research areas.

In addition to testing the CEIRS network response capability, the authors provide a wealth of data on emerging canine influenza viruses (CIV) as a pandemic threat. In this area the authors test the transmissibility of CIV in multiple animal models, characterize the phenotype of these viruses, and assess pre-existing immunity in the human population by age. Below are specific comments for each of the parameters tested of CIV that would assist the reader in following this comprehensive study.

**Part II – Major Issues: Key Experiments Required for Acceptance**

Reviewer #1: - Many of the experimental efforts did not include a representative H3N8 virus. Is there a reason for that? Without this, it is impossible to determine how broadly the results in question extend to all canine H3 viruses or just H3N2s.

- Figure 12 is missing

- The ferret transmission experiment uses a physiologically irrelevant infection dose of 10^6 pfu. I recognize that the flu field has settled on this dose as the gold standard for ferret infections for some reason, but it makes it impossible to determine whether the transmission results in that experiment reflect what would be expected during natural infection. This experiment should either be repeated using a relevant dose (as was done for the guinea pig experiments), removed form the paper, or the text should clearly highlight that the results may be non-representative of reality due to the experimental design.

Reviewer #2: The panel of viruses examined included natural H3N2 viruses from dogs and their RG equivalents, an H3N8 virus from dogs and two RGs based on it, along with a panel of human H3N2 viruses and some RG reassortants. The passage histories of the viruses have not been given and, for human H3N2 viruses at least, passage history and the changes acquired on passage need to be taken into serious consideration. Whether this is the case for canine H3 viruses ought also to be considered.

Table 3 shows the low pH inactivation of virus and the pH of syncytium formation by three viruses. The authors need to present a conclusion on whether the differences observed between the pH of inactivation of the two canine viruses is significantly different from the single example of a human H3N2 virus. The conclusion presented on lines 193-194 seems to require further justification.

The authors describe pathogenicity studies in mice and show that the DBA/2J mice are sensitive to infection by CIV-41915 and rCIV-1177. It might be worth explaining whether CIV-41915 and rCIV-1177 resulted in the death of those mice scored as fatally infected or whether they were culled having reached a humane end-point. It Is not clear what conclusion ought to be reached with such varying results between inbred mouse lines.

Figure 6 shows the results of transmission between infected ferrets or guinea pigs and direct contacts. The style of the results being presented ought to be generally similar. Figure 5A does not seem to look to be data from nine single animals (as described in the materials and methods (lines 633 to 634) and here it is implied that more than one virus was tested for ferrets).

Table 5 and Figure 6 show antiviral sensitivity results. Here different viruses were assessed by different methods. It is particularly striking that the absolute sensitivity of the different viruses to the drugs are different in the different assays. This is not perhaps surprising for the NA inhibitors (a sialidase inhibition assay versus a virus replication assay) but the assays for amantadine resistance have similar types of read-outs and the sensitivity of the viruses to amantadine appear to be very different indeed. That said, there is no doubt that the conclusions drawn overall are robust.

Antigenic differences between the viruses and circulating human viruses have been assessed in several ways. Amino acid sequence-based methods have been presented here extensively based on the claim (lines 246 to 248) that these methods can predict antigenic relatedness. Figure 7 shows a cluster analysis of the viruses but this does not directly assess the antigenic relatedness of the viruses within a subtype. No assessment of antigenic discrimination by post-infection ferret antisera has been given – yet the sera might well have been collected in the studies. With these sera the hypotheses presented in the paragraph from lines 271 to 280 (particularly lines 278 to 280) about the relatedness of the CIV viruses to old H3N2 viruses ought to have been tested directly.

Human immunity has been subsequently analysed. It is very surprising that there are such high titres seen to the A/Wyoming/3/2003 antigen by HI. I would presume that these sera would be considered to be the equivalent of pre-vaccination sera and a typical panel of pre-vaccination human sera might have mean GMTs of 30 to 100. These HI titres presented here for A/Wyoming/3/2003 look very much higher with some antisera having titres of over 1,000 and so the protocols used need to be reassessed by the panel of authors. With the HI panel showing such a concern for A/Wyoming/3/2003, all the other results also need to be carefully reassessed. 

Human post-vaccination sera were also analysed with two panels of sera. The authors describe a ‘baseline’ titre of neutralising antibody – I have presumed that this is pre-vaccination titres. It is a pity that these were not also assessed by HI assay as well as the VN assay. The results shown in figures 8B and 8C seem to show pre-vaccination and post-vaccination titres in the figure legend, but this is not what is shown in the figure which shows unvaccinated and vaccinated individuals. I was confused which was the correct interpretation. Pre- and post-vaccination paired sera would present the strongest evidence for any cross-reactivity or not.

Human mAbs were also analysed. Here a large panel of mAbs have been put into groups but the epitopes recognised by the panel of mAbs do not seem to have been defined and this makes interpretation of the results difficult. This also applies to Figure 11 – and the coding had to be deduced from the results. These results were also very difficult to interpret. Table 6 is not well described or well presented. The significance of the results in this whole section is difficult to establish.

One of the key elements of WHO’s TIPRA is for the experts to make an assessment of the confidence scores for each of the scoring elements. Here this does not seem to have been carried out; this could well have been done either in the results section or, perhaps better, in the discussion. This assessment would allow the authors to assess the significance of their own results. An important point of the risk assessment frameworks is to assess where there is limited information, thereby indicating what types of new data are required for improved risk assessments. The authors should take an opportunity to do this. They might cover the points raised here about difficulty in interpretation of some of the presented results.

The discussion, as it stands, repeats many of the points covered in the introduction. However, the discussion raises other very important points – particularly relating to the use of RG viruses. The authors offer a possible solution to this.

Reviewer #3: (No Response)

**Part III – Minor Issues: Editorial and Data Presentation Modifications**

Reviewer #1: - Rationale for the specific virus strain selections in fig 1 and fig 2 not well explained

- Line 158: data do not actually show a comparison between natural and recombinant versions of 11613, the natural isolate tested is 41915.

- Fig 2: no description of 2E anywhere, and 2D is not described in the legend

- Fig 3: x-axes would help. Also, the examination of a single canine isolate seems insufficient to draw broad conclusions. should compare representatives from H2N8 and H3N2 trees

- Line 279: Does this analysis provide any information not provided by a maximum likelihood phylogenetic tree of the relevant gene segments?

- Figure 7: For consistency, would be good to indicate the specific strains used in the human serological analyses in figure 8 on these plots.

- The high, age-independent ELISA and ELLA titers against the canine N2 and N8 proteins are surprising, as the authors note. Does this suggest that the sequence-based antigenic analysis fails to accurately capture something important about NA-Ab interactions?

- Figure 10E: legend is insufficient. What are values shown on heat map? Also, do the IF data suggest that AIR may miss some cross-reactivity? For example, 017-10116-5B03 shows maximal signal against rWY03 by IF but looks like it has no signal against this virus by AIR? It looks like the discordance between the two datasets is more complicated than simply a greater degree of sensitivity of AIR vs. IF as the authors claim. For example, 028-10134-4F03 apparently shows strong reactivity with rCIV-11613, moderate reactivity with rWY03 and little to no reactivity with rCIV-23 and HK68 by IF in figure 11, but the same mAb shows significant reactivity with the two canine IAVs but not HK68 or rWY03 by AIR in fig. 10. This is a qualitative difference in the predicted reactivity, not simply a difference in sensitivity. There may be others as well, but I don’t have time to scan through the whole thing. The authors should confirm whether the differences in results provided by the two methods are a consistent sensitivity issue or are more complicated.

Reviewer #2: Minor points

The manuscript has the signs of being written by several authors. I would suggest a single author goes through the whole manuscript (particularly in the materials and methods section) to ensure a consistent standard of use of English and the amount of detail given.

Line 31. The words ‘minimal risk’ are probably not the appropriate words to describe the outcome since ‘low risk’ would be better. It is easy to think of influenza viruses of lower risk to humans than the mammalian adapted canine influenza viruses.

Line 90. If the authors mean Low Pathogenicity (or Low Pathogenic) Avian Influenza viruses then this should be clearly stated as it has a precise meaning in OIE regulations and other legislation, whereas nonpathogenic is not well defined.

Line 96. It would be nice to provide a few more words about the nature of the human influenza viruses isolated from dogs and how well they were characterised.

Lines 201 and 211. PFU units were used for the infection of guinea pigs and TCID50 units were used for infecting ferrets. It would be good if the authors were able to equate the infecting doses.

Line 205. To reduce the possibility of confusion, I think the term ‘line’ should be used for the mice rather than the word ‘strain’, which might be taken to refer to a virus.

Lines 278 and 321 and Table 2 – I suspect the authors might be referring to A/Hong Kong/4801/2014 since this was presumably the vaccine virus used (line 321) rather than A/Hong Kong/1/2014.

Line 537. A reference to TEC MM is needed.

Line 540. I suspect the word ‘residue’ might well ought to have been ‘residual’.

Lines 592-593, the nature of the MDCK Luc9.1 cell needs to be given.

Reviewer #3: Specific comments:

Addition of a timeline figure to visually demonstrate the length each component took the contribution of each CEIRS centers and highlight areas where the networks worked together would be very useful for this exercise. Based on the discussion (line 430-431) it seems that all of this work was completed within 2 months – which is outstanding for a first test and highlights the power of the CEIRS network.

List viruses used in study

• Please add the passage history for both natural isolates and recombinant viruses to this list, as it may be distinct for each virus and important to understand if they were passaged in eggs prior to analysis for viral phenotypic characteristics. Especially since these details are not provided in the materials and methods section.

Replication in primary human respiratory epithelial cells and explants:

• CIV-1177 is denoted at recombinant in the text (line 163) but not in figure 2 or line 168. Given that both recombinant and non-recombinant strains were used, it is necessary to ensure that each is denoted accurately for the reader.

Transmission in ferrets and Guinea pigs

• Line 216-217 states there was ‘no airborne transmission’ of CIV-41915 yet one animal shed detectable virus on day 3 post infection. Could the authors please provide the serology for this airborne transmission study as well as the one for rCIV-1177 which is listed as data not shown.

Cross-reactive human and CIV antibodies

1. Figure 8 and 9 are very well done and present exciting observation critical to address pandemic emergence and the proportion of the population that is susceptible to CIV.

2. The use of AIR is important – but the data presented in figure 10 is difficult to interrupt for the general reader. The majority of the antibodies tested are not specific to any virus and overwhelm part E. In addition it is unclear how A-D relate to the antibodies presented in figure E. Please adjust this presentation of these data to make it more reader-friendly. Also, a different nomenclature for the antibodies might be useful. A minor note is that some rows do not line up on the left edge of the heatmap – which is surprising and this reviewer is left wondering whether the heatmaps were generated at multiple times and then spliced and fit together? It seems as though all the data should be imported into the same script for heatmap generation.

3. Could the authors please denote them in figure 10 so the reader can easily compare the AIR profile to the cross-reactive IF profile. Even though it is presented on Table 6 – it would be nice to assess it in the figures. On Table 6 – could the authors add dividing lines between groups and antibodies.

In vitro neutralization of H3 CIVs with hmMAbs

• Figure 12 was not included in the manuscript so nothing could be assessed. Although the NT titers are provided in Table 6 – suggesting that Figure 12 is unnecessary.

A final comment is that there are a number of grammatical typos that should be fixed. Some examples include lines 94 (their -> the), 106 (add in ‘to’ between obtained and inform), 348 (30 should come after ‘of the’).

PLOS authors have the option to publish the peer review history of their article (what does this mean?). If published, this will include your full peer review and any attached files.

Reviewer #1: No

Reviewer #2: No

Reviewer #3: No

---

## [Editor Report · Decision Letter 1]

6 Feb 2020

Dear Dr. Schultz-Cherry,

Thank you very much for submitting your manuscript "Characterizing Emerging Canine H3 Influenza Viruses." for consideration at PLOS Pathogens. Your revised manuscript was reviewed by members of the editorial board.  We are likely to accept this manuscript for publication, providing that you modify the manuscript as indicated below:

1. The phrase "the CEIRS network could work together..." is repeated at least three times in the first few pages of the manuscript (in the abstract, last paragraph of the intro, and first paragraph of the results section). Please remove this sentence from the abstract and introduction as it distracts from the scientific rationale of the work.

2. In the abstract, please either expand on the "valuable lessons learned" (last sentence) or remove this sentence from the abstract as it is vague.

3. Unites States is define as US in line 45, but this definition is not used in the remaining of the manuscript.

4. line 58: A/Hong/Kong... please add influenza A virus.

5. line 78: "preexisting antibody (Ab) immunity"...that wording is not conventional use preexisting antibodies or preexisting humoral immunity.

Sincerely,

Carolina B. Lopez, Ph.D.

Associate Editor

PLOS Pathogens

Volker Thiel

Section Editor

PLOS Pathogens

Kasturi Haldar

Editor-in-Chief

PLOS Pathogens

orcid.org/0000-0001-5065-158X

Michael Malim

Editor-in-Chief

PLOS Pathogens

orcid.org/0000-0002-7699-2064
---

## [Editor Report · Decision Letter 2]

19 Feb 2020

Dear Dr. Schultz-Cherry,

We are pleased to inform you that your manuscript 'Characterizing Emerging Canine H3 Influenza Viruses.' has been provisionally accepted for publication in PLOS Pathogens.

Before your manuscript can be formally accepted you will need to complete some formatting changes, which you will receive in a follow up email. A member of our team will be in touch within two working days with a set of requests.

Best regards,

Carolina B. Lopez, Ph.D.

Associate Editor

PLOS Pathogens

Volker Thiel

Section Editor

PLOS Pathogens

Kasturi Haldar

Editor-in-Chief

PLOS Pathogens

orcid.org/0000-0001-5065-158X

Michael Malim

Editor-in-Chief

PLOS Pathogens

orcid.org/0000-0002-7699-2064
---

## [Editor Report · Acceptance letter]

8 Apr 2020

Dear Dr. Schultz-Cherry,

We are delighted to inform you that your manuscript, "Characterizing Emerging Canine H3 Influenza Viruses.," has been formally accepted for publication in PLOS Pathogens.

Best regards,

Kasturi Haldar

Editor-in-Chief

PLOS Pathogens

orcid.org/0000-0001-5065-158X

Michael Malim

Editor-in-Chief

PLOS Pathogens

orcid.org/0000-0002-7699-2064